# Dendritic heterosynaptic plasticity arises from calcium-based input learning
Shirin Shafiee [1,2] ✉, Sebastian Schmitt [1,2] & Christian Tetzlaff [1,2]

Stimulus-triggered synaptic plasticity is the foundation of learning and crucial cognitive abilities. Although numerous computational models have investigated plasticity within networks of point neurons, dendritic integration confers superior computational capacity compared to these simplistic models, highlighting the significance of dendrites and their spines–small, specialized protrusions that serve as loci for synaptic plasticity. Synaptic plasticity can be categorized into two forms: homosynaptic plasticity, involving changes at directly stimulated synapses, and heterosynaptic plasticity, involving changes at non-stimulated synapses. For homosynaptic plasticity, the $Ca^{2+}$-hypothesis identifies the calcium concentration within a stimulated dendritic spine as the key mediator. In contrast, although theoretical studies attribute important roles such as synaptic competition and cooperation to heterosynaptic plasticity, experimental evidence remains ambiguous. By integrating insights from $Ca^{2+}$-dependent homosynaptic plasticity with data on dendritic $Ca^{2+}$-dynamics, we demonstrate that calcium influx into a stimulated spine can diffuse to neighboring spines, triggering heterosynaptic effects. To investigate this, we develop a mathematical model characterizing the temporal and spatial dynamics of calcium in dendrites in response to different inputs. Our model explains experimental ambiguities and extends the $Ca^{2+}$-hypothesis to heterosynaptic plasticity. Notably, it predicts that input-timing, distance between spines, and local diffusion properties modulate synaptic changes, revealing a mechanism for dendritic computation.

Long-term synaptic plasticity is a crucial mechanism for learning and memory in the brain[1,2] and inspiration for modern artificial neural network algorithms[3,4], with the majority of studies focusing on homosynaptic plasticity, which involves modifications confined to the directly activated synapses or dendritic spines. On the other hand, experimental studies have revealed that the induction of synaptic plasticity exerts its influence not only at the synapse located on the stimulated spine, but also at neighboring spines[5–7]. As shown by theoretical studies, such heterosynaptic influences are required for network stabilization or synaptic competition[6,8–10]. Unlike homosynaptic plasticity, the biological underpinnings of heterosynaptic plasticity and its detailed implementation in models remain largely unknown[11].

Homosynaptic plasticity research has centered on the idea that intracellular calcium concentration in dendritic spines is a key regulator of synaptic strength, commonly referred to as the $Ca^{2+}$-hypothesis. This framework posits that the level of intracellular calcium ($[Ca^{2+}]$) determines the direction of synaptic plasticity: high $[Ca^{2+}]$ induces long-term potentiation (LTP), moderate levels trigger long-term depression (LTD), and low levels produce little or no synaptic change[12–17]. In this context, calcium serves primarily as a secondary messenger, initiating and regulating signaling cascades through enzymes, such as phosphatases and kinases, that mediate synaptic plasticity[14].

In recent years, studies have shown that dendrites hosting spines have a more active role than a passive cable that sums synaptic signals and transmits them to the soma[18,19]. For instance, dendrites exhibit nonlinear integration of electrical signals[20,21] and can detect sequences of inputs[22,23]. As a result, dendrites are considered biochemical units that play a crucial role in synaptic plasticity[24,25]. Moreover, in-vitro and in-vivo experiments have demonstrated that correlated synaptic inputs cause an elevation in calcium concentration in the dendrite[13,26–30], and strong inputs can even lead to the release of $Ca^{2+}$ from local dendritic stores[31–36].

In this study, we developed a computational model integrating the well-established $Ca^{2+}$-hypothesis of synaptic plasticity with recent findings on dendritic $Ca^{2+}$ dynamics. We demonstrate that, upon stimulation, elevated $Ca^{2+}$ concentration in one spine can diffuse through the dendritic shaft to neighboring spines. Depending on the state of the neighboring spine, this diffused calcium can produce substantial changes in homo- and heterosynaptic plasticity, resulting in a potential learning rule in which

[1]III. Institute of Physics-Biophysics, Faculty of Physics, University of Göttingen, Göttingen, Germany. [2]Group of Computational Synaptic Physiology, Department of Neuro- and Sensory Physiology, University Medical Center Göttingen, Göttingen, Germany. ✉e-mail: shirin.shafieekamalabad@uni-goettingen.de

synaptic changes depend on the relative timing of synaptic inputs to neighboring spines, without requiring postsynaptic action potentials. To explore the resulting complex Ca$^{2+}$ and plasticity dynamics, we systematically varied the stimulation protocol, the number of spines, and Ca$^{2+}$-related parameters of both spines and the dendrite. First, in a system of two spines connected by a dendrite, we demonstrate the emergence of hetero-synaptic competition and cooperation depending on input frequency. Subsequently, we modeled larger systems with several spines along a dendritic shaft, using protocols from multiple experimental datasets[37–40] to reproduce their key results across different input frequencies. Our findings support the idea that Ca$^{2+}$ diffusion plays a crucial role in regulating heterosynaptic plasticity.

Considering a two-spine system, we demonstrate that calcium-based homo- and heterosynaptic plasticity can yield input-timing-sensitive synaptic changes. Notably, these changes occur under the sub-threshold regime of dendritic integration, where no postsynaptic action potentials are generated. In contrast to the traditional spike-timing-dependent plasticity (STDP) protocol, which relies on the timing of pre- and postsynaptic events to determine synaptic weight changes, our results highlight the crucial role of inter-input timing in this context. Similar to the well-established STDP protocols for homosynaptic plasticity[14,41], we observe complex "temporal windows" of potentiation and depression, providing a basis for learning rules that underscore the potential role of heterosynaptic plasticity in computation and cognition.

Furthermore, we propose a simplified model of calcium diffusion in which calcium entry can occur through various candidate channels and receptors, such as NMDARs or VDCCs. Our goal is not to identify the precise channel(s) responsible. Although some studies suggest that NMDARs act as coincidence detectors and gates for calcium entry, other studies indicate that synaptic plasticity can occur independently of NMDAR coincidence detection. This suggests that alternative mechanisms–such as voltage-gated calcium channels, AMPARs, or neuromodulators–can drive synaptic changes when NMDARs lose their Mg$^{2+}$ block[42]. Non-NMDA-dependent LTD has been experimentally observed in certain brain regions[43]. For example, AMPA receptors and VGCCs can induce LTD at cerebellar parallel fiber (PF) synapses onto Purkinje cells[44].

In the barrel cortex, experience-dependent plasticity can occur via calcium-permeable AMPARs (CP-AMPARs) without NMDAR involvement. Ca$^{2+}$-permeable AMPARs at layer 4-layer 2/3 synapses can modulate plasticity by providing an alternative source of Ca$^{2+}$ influx, potentially supporting NMDAR-independent synaptic modifications[45]. While NMDARs are central to many forms of synaptic plasticity (especially Hebbian LTP/LTD), these studies demonstrate that alternative mechanisms —such as mGluR signaling, VGCCs, dopamine modulation, BDNF, and CP-AMPARs—can mediate plasticity in specific brain regions or under certain conditions[43,46,47]. Thus, since NMDARs are not always required for plasticity, and their necessity depends on the synaptic pathway and learning paradigm, we model extracellular calcium entry through a representative channel.

Notably, some findings reveal that dendrites allow different plasticity rules to dominate in distinct regions of the neuron, influencing how connections between cells strengthen or weaken[48]. Furthermore, the canonical STDP rule has limited learning capacity in the case of weak or spatially distal inputs. Intriguingly, neurons incapable of generating strong back-propagating action potentials may instead utilize localized subthreshold activity as a plasticity substrate[48,49]. Remarkably, such subthreshold depolarizations can elicit LTP and LTD despite identical pre- and postsynaptic spike patterns[50]. Hence, we propose a learning mechanism based on local subthreshold events and calcium-dependent input timing.

## Results

Using our computational model of the dynamics of freely diffusing Ca$^{2+}$ ions (Fig. 1), we first considered a piece of dendrite, at which two similar spines were situated with a distance of 1 μm. We injected a synaptic current at one of these two spines (spine 1, blue in Fig. 2A), while the other spine remained unstimulated (spine 2, orange). Synaptic activity resulted in a high calcium concentration and an accumulation of calcium ions at the stimulated spine (Fig. 2A, middle), leading to potentiation (right). Subsequently, free calcium ions diffused to adjacent dendritic compartments and, consequently, to the non-stimulated spine. Here, diffusion yielded a mildly increased calcium level in spine 2 such that the non-stimulated synapse underwent a slight depression (orange, Fig. 2A, right). This pattern of

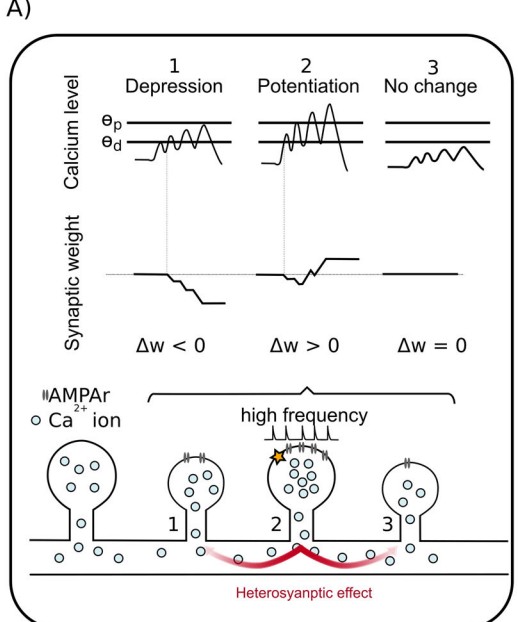

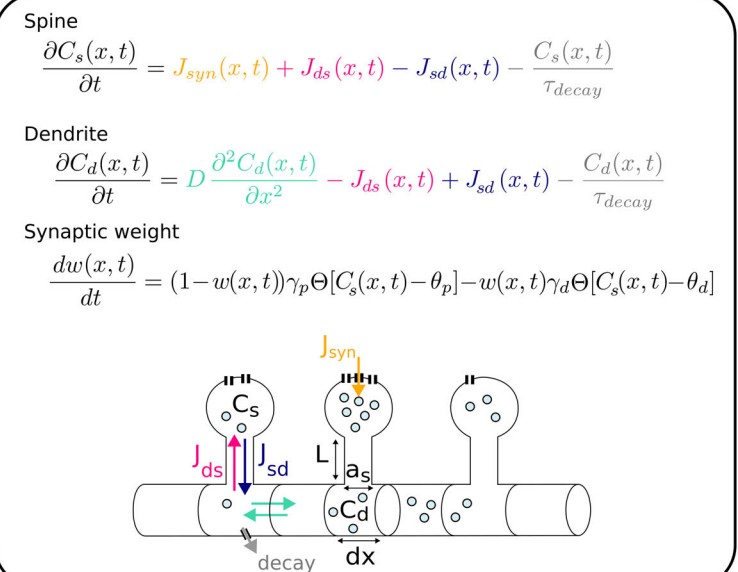

**Fig. 1 | Illustration of our model of calcium diffusion between spines. A** Schematic of the main mechanisms in our model, given the example of three neighboring spines: the stimulated spine (#2) experiences potentiation, while the calcium level of spine #1 could only rise slightly and undergoes depression. A modest amount of calcium in spine #3 does not substantially alter the synaptic weight. **B** Mathematical model of the calcium dynamics and resulting synaptic weight changes. For more details, see the "Methods" section.

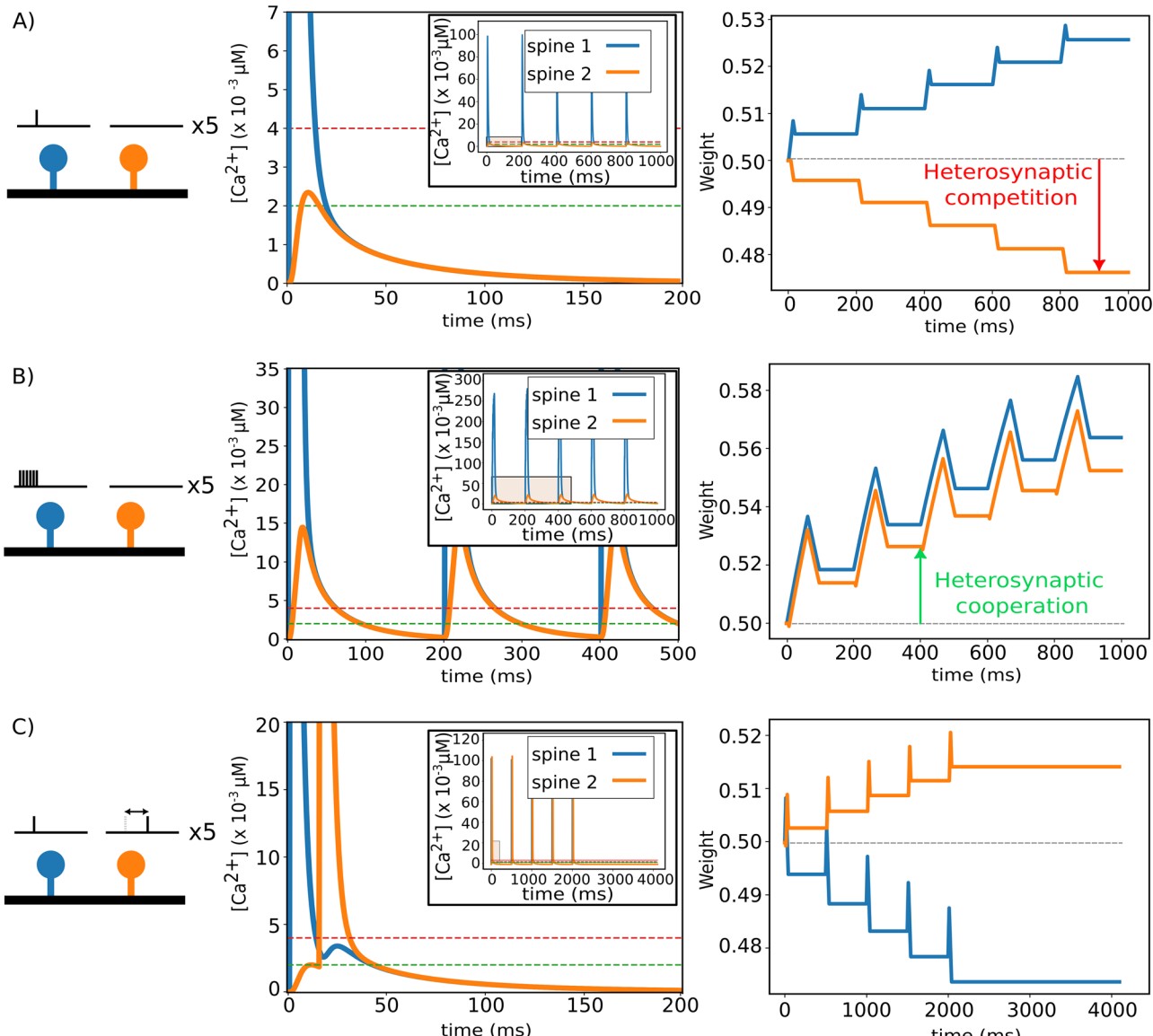

**Fig. 2 | Heterosynaptic effects of different stimulation protocols.** Each stimulation protocol is given 5 times. Left: stimulation protocol; middle: temporal development of calcium concentrations in spines; right: temporal development of synaptic weights. Blue: spine 1; orange: spine 2. **A** Single input stimulation at one spine, while the second spine receives no input signal. **B** Burst of inputs at one spine, with the second spine being unstimulated. **C** Single input stimulation of both spines with time difference of 15 ms. Parameters: $\theta_d = 2.10^{-3}$ µM (green dash line), $\theta_p = 4.10^{-3}$ µM (red dash line).

homo- (spine 1) and heterosynaptic (spine 2) plasticity is referred to as synaptic competition[5,6,40]. Figure 2B shows the influence of high-frequency stimulation on the magnitude and direction of plasticity. In contrast to a single spike stimulation, subjecting spine 1 to a burst of stimulations caused a more pronounced elevation of calcium level at the neighboring spine located in proximity to the stimulated spine (orange, Fig. 2B). This elevated calcium concentration resulted in potentiation of both the stimulated spine and the adjacent unstimulated spine, introducing heterosynaptic cooperation[5,51]. Figure 2C illustrates another protocol similar to the first one in Fig. 2A. We initially stimulated the first spine (blue), but different to the previous protocol, here, this stimulation was followed by a stimulation of the second spine (orange) 15 ms later. Compared to the first protocol, stimulation of the second spine completely changed the pattern of plasticity, triggering synaptic depression at spine 1 and potentiation of spine 2 (Fig. 2C, right). To better illustrate the impact of spike timing on synaptic plasticity, we intentionally adjusted the potentiation and depression thresholds in Fig. 2 to be lower than those in Fig. 3, thereby amplifying the visibility of subtle timing-dependent changes.

Next, we extended our model to consider a dendrite with multiple spines in order to compare our model results with various experimental datasets[38–40]. Experimentally reported plasticity patterns after stimulation show considerable variability and lack a coherent explanation.

To integrate all three datasets, we modeled an 80 µm-long dendrite hosting 16 randomly distributed spines governed by calcium-dependent synaptic plasticity (Fig. 3A). All protocols were repeated for 20 different random spine configurations, with initial synaptic weights randomly chosen to represent different initial spine sizes and volumes observed experimentally. A random subset of 7 spines received stimulation and formed the group of stimulated spines (Stim, teal color). We ensured that at least three non-stimulated spines were located outside the group of stimulated spines on each side (Un$_{out}$, pastel yellow). The remaining spines were non-stimulated and formed the group of inner neighboring spines (Un$_{in}$, coral pink). Note that the spatial distribution of spines is arranged such that a total of 10 spines, comprising 7 stimulated and 3 unstimulated ones, are confined to a 7 µm section of a dendritic branch, yielding a spine density of approximately 1.4 spines/µm, following reported experimental values[40,52–54].

**Fig. 3 | Heterosynaptic effects of different stimu-lation protocols. A** Arrangement of stimulated (in teal color) nearby unstimulated (in coral pink), and distant unstimulated spines (in pastel yellow). **B–D** Simulation results with low ($D = 1 \frac{\mu m^2}{s}$) and physiological diffusion constant ($D = 220 \frac{\mu m^2}{s}$) as well as experimental results. **B** Simulation results with low input frequency as well as experimental observation (see Fig. 1 of Oh et al.[37]). **C** Simulation results at medium frequency and experimental observation (see Fig. 2c-iii of Tong et al.[39]), and **D** simulation at high frequency as well as experimental observation (see Supplementary Fig. 3 of Chater et al.[40]). Each data point corresponds to the synaptic weight change of a specific synapse type (Stim, Un_in, or Un_out) obtained after simulation at a particular frequency and with a specific spatial arrangement of spines. The standard deviation reflects the variability in synaptic weight changes obtained across different spatial arrangements of spines. Error bars represent the standard deviation of synaptic weight changes across 20 different spatial arrangements of spines. **E** Errors between simula-tion and experiments shown in (**B–D**) for different diffusion constants (see synaptic weight patterns in Supplementary Fig. 8). Black data points represent the total error for a specific spine arrangement across all frequencies and spine groups. Note that diffusion constants are plotted in logarithmic scale. The standard deviation represents the variability in error values obtained from different spatial arrangements of spines.

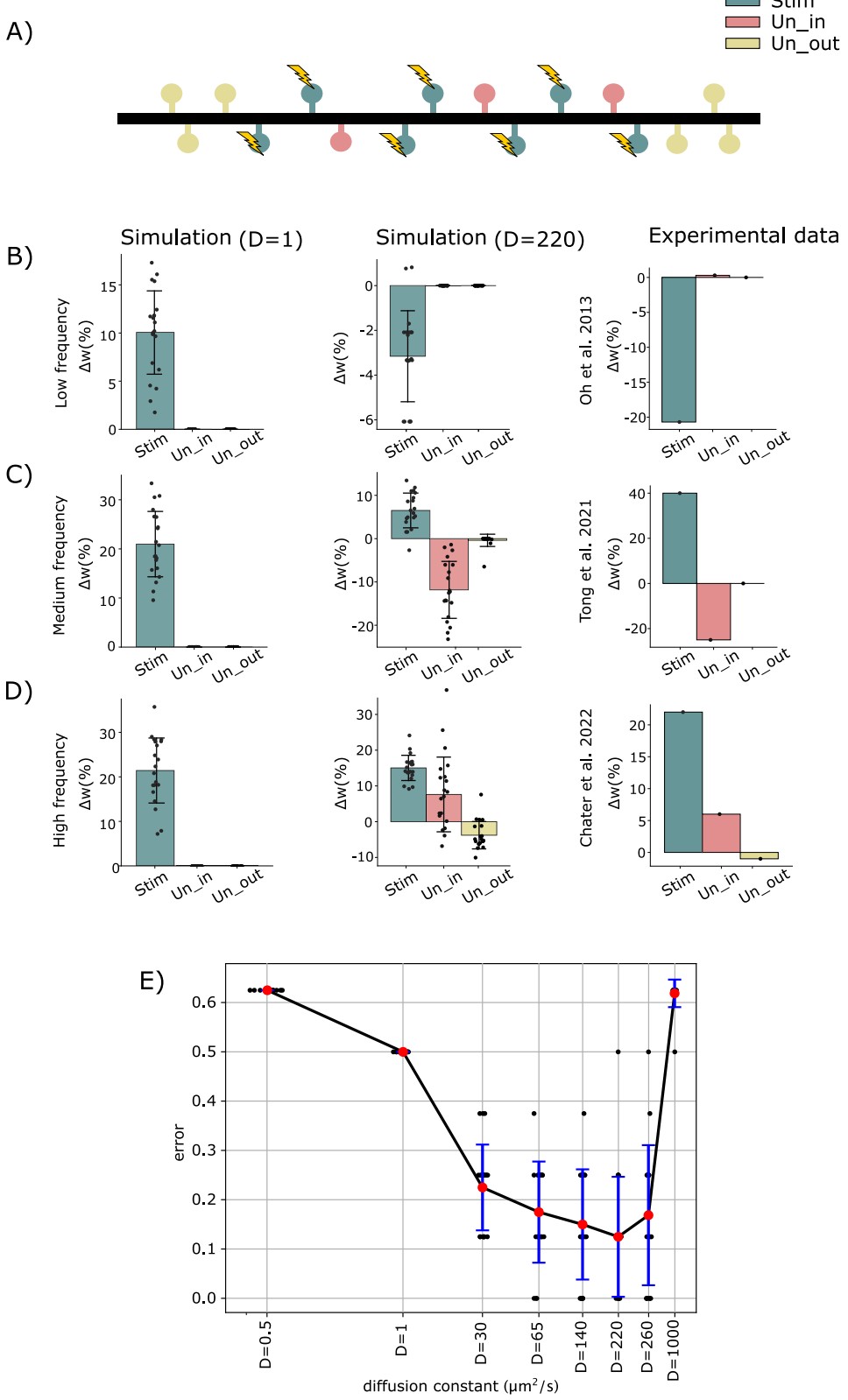

It was shown that low-frequency uncaging induces synapse-specific spine shrinkage and, thus, likely LTD, while unstimulated synapses remain unaffected[37] (Fig. 3B). To match this pattern of plasticity, we applied low-frequency stimulation to 7 of the 16 randomly distributed spines and observed, on average, a depression of synaptic weights in the Stim group. Consistent with experimental findings, unstimulated spines maintained their initial state, showing no significant plasticity.

If we increase the stimulation frequency of the targeted stimulated spines, we can qualitatively match the pattern of plasticity from Tong et al.[39] (Fig. 3C). Here, the average synaptic weights of stimulated spines are potentiated, while we observe LTD at the inner neighboring, unstimulated spines and no significant change at the outer, unstimulated spines.

Further increasing the stimulation frequency at targeted spines enables us to match the experimental results reported by Chater et al.[40] (Fig. 3D).

Using glutamate uncaging, the authors observed that, after stimulation, the volumes (and accordingly the synaptic weights) of stimulated and inner neighboring, unstimulated spines grow, while the volume of the $Un_{out}$ group decreases.

Note that for all three protocols, we show the pattern of plasticity averaged across 20 different spine distributions. For individual distributions, results could vary considerably in the model as well as in experiments (see Supplementary Fig. 7 and also ref. 40). Furthermore, the induction of heterosynaptic plasticity is essential for all protocols. If we reduce the effective diffusion constant such that heterosynaptic plasticity will have a negligible effect on the plasticity pattern, model results always show only homosynaptic potentiation and no plasticity at unstimulated spines (left, Fig. 3B–D).

We repeated all protocols with different values of the diffusion constant ($D = 1, 30, 65, 140, 220, 260, 300, 350, 400, 440 \frac{\mu m^2}{s}$) and 20 random configurations each (Fig. 3E). Then, we counted the number of instances where the resulting plasticity patterns matched the plasticity pattern of the corresponding experiment for each spine group (i.e., Stim, $Un_{in}$, and $Un_{out}$, as shown in the last column of Fig. 3B–D), and derived a matching error between 0 and 1. It is evident from Fig. 3E that the lowest error value corresponds to a range of diffusion constants near the calcium diffusion constant. Although reducing the diffusion constant in our model diminishes the heterosynaptic effect by design. However, it is worth noting that this change not only impairs heterosynaptic plasticity at the corresponding frequency but also affects the homosynaptic pattern, which no longer matches the reported experimental results. For instance, in Fig. 3B–D (left panel), stimulated spines undergo potentiation at $D = 1$, whereas at $D = 220$ they undergo depression, consistent with experimental observations. This result supports the model assumption of calcium being one of the main candidates to communicate heterosynaptic plasticity.

After matching the model to experimental data, we next analyzed the influence of input timing on the plasticity pattern of a system consisting of two spines. Similar to homosynaptic STDP, we evaluated the temporal window of plasticity by considering the temporal difference between a spike at spine 1 and a spike at spine 2 and the resulting synaptic weight changes. Note that we obtained two curves, as two synapses are always involved. Fig. 4A shows the categorized patterns of synaptic weights at different input time differences. It is evident that increasing $\gamma_d$ enhances the level of depression, thereby shifting the temporal window towards depression for all time differences (green region in Fig. 4A). Similarly, increasing the potentiation coefficient, $\gamma_p$, shifts the temporal window towards potentiation for all time differences (red region in Fig. 4A). There are intermediate values of $\gamma_d$ and $\gamma_p$ at which the temporal windows become more complex.

It is noteworthy that the majority of existing experimental and theoretical studies have primarily focused on the pair activity of neurons[55,56] and external calcium[56]. Consequently, the parameter values corresponding to $\theta_{p/d}$ and $\gamma_{p/d}$ cannot be directly compared to these studies. Furthermore, previous work, such as Inglebert et al.[56], which employed parameter fitting, still suggests multiple ranges of values for some of these parameters. Other computational studies[14,15] have also focused on pairing protocols and proposed different values for these parameters. Although we explored a wide range of $\theta_{p/d}$ and $\gamma_{p/d}$ and selected values that match different experimental observations, a fair comparison and quantification of the parameters of our model necessitate a compatible experimental setup.

In Fig. 4B, we varied the distance between two spines (1, 2, and 3 μm) and performed the same time-difference protocol. An increased inter-spine distance correlates with a diminished capacity for one spine to influence the state of its neighbors. Consequently, spines situated further apart exhibit reduced heterosynaptic plasticity, tending towards an isolated state (also see Supplementary Fig. 2).

It has been shown that traditional pair-based STDP models fail to account for complex temporal interactions or explain experimental observations. Therefore, a triplet protocol was proposed to overcome the limitations of canonical pair-based homosynaptic STDP models[57,58]. Following this idea, we employed a triplet stimulation protocol involving two input

spikes at one spine and a single input spike at a second spine (Fig. 4C). Time differences were measured relative to the arrival of the first input spike at spine 1. As anticipated from a comparison of Fig. 4A, C, introducing a third input spike significantly increases the complexity and diversity of the temporal windows of plasticity.

## Temporal sequence selectivity

On the one hand, as shown in Figs. 2 and 3, dendritic branches are sensitive to the timing and location of synaptic inputs at the spines. Depending on these input features, calcium-based learning can result in distinct signaling patterns within a branchlet. On the other hand, experimental evidence indicates that dendritic branches enable the soma to discriminate the direction of the sequence of inputs—whether it is from the tip of the branch towards the soma (inward sequence), or vice versa as an outward sequence (see Branco et al.[23] and Fig. 5A). A passive dendritic cable without synaptic plasticity will always result in the soma to show the maximum response to the inward sequence. However, for a neuron to be able to discriminate arbitrary input sequences, it should be able to learn to maximally respond to outward sequences as well.

To show that calcium-diffusion-dependent plasticity enables a neuron to learn to discriminate between inward and outward sequences, we introduced a somatic compartment positioned on one end of the dendrite. Hereby, a current flows toward the soma when the membrane potential at the terminal dendritic compartment exceeds the resting potential of the soma. Notably, we did not account for feedback from the soma to the dendrite, neglecting backpropagating effects. The somatic compartment is modeled as a simple integrative unit connected to the terminal dendritic compartment via an axial resistance, $R_{s,axial}$. Its electrical properties are defined by its resistance ($R_{soma}$) and capacitance ($C_{soma}$). Here, $u_{rest} = -70$ mV, $\tau = R_{soma}C_{soma} = 22$ ms[59] and $R_{soma}/R_{s,axial} = 5$. Note that, $u_{dend}(x_0, t)$ represents the voltage at the dendritic compartment connected to the somatic compartment.

$$\frac{du_{soma}(t)}{dt} = \frac{1}{\tau}\left[(u_{rest} - u_{soma}(t)) + \frac{R_{soma}(u_{dend}(x_0, t) - u_{soma}(t))}{R_{s,axial}}\right]. \quad (1)$$

For simplicity, we considered two dendritic spines receiving synaptic inputs through two synapses. These spines were positioned 1 μm apart and located proximally to the soma (5 μm away).

We first presented the inward input sequence five times to the dendrite (i.e., $t_2 < t_1$) during the learning phase. After the learning phase, we presented the inward and outward sequences once, separately, and recorded the somatic membrane potential. We observed that the soma reached its peak potential when the inward pattern was presented (Fig. 5B). In contrast to a passive, nonplastic dendrite presenting the outward pattern during the learning phase, resulted in the soma exhibiting its maximum response when the outward pattern was presented during the test phase. These findings demonstrate that, by incorporating diffusion-dependent calcium-based synaptic plasticity, our model of a simple dendritic branch can learn different sequences of input signals.

## Discussion and conclusion

We developed a computational model of molecular diffusion dynamics in a piece of dendrite and connected spines. Using a diffusion constant similar to that of calcium, our model can match results from several experimental studies of homo- and heterosynaptic plasticity. Our model also demonstrates that, given the sensitivity of calcium influx to the timing of presynaptic input spikes, the triggered homo- and heterosynaptic plasticity can lead to complex input-STDP patterns. This provides a rich repertoire of calcium-based learning rules without the need for postsynaptic spiking.

Previous experimental studies have reported seemingly contradictory results[11,38,39], yet a comprehensive mechanism explaining these discrepancies has not been proposed. Chater et al.[40] investigated structural long-term

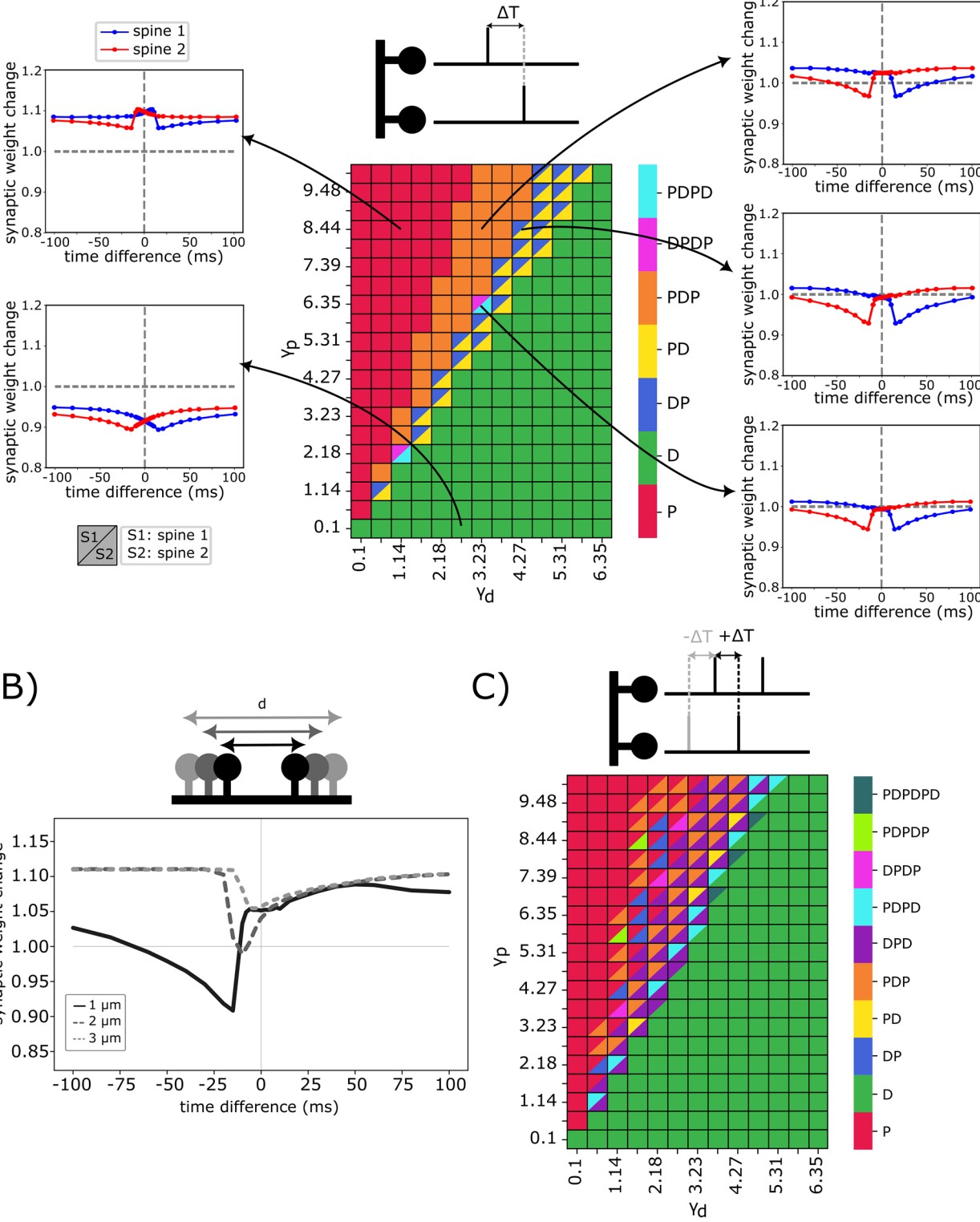

**Fig. 4 | Various curves of heterosynaptic, input-timing plasticity. A** Emergent temporal windows after stimulation of two spines at various time differences. Both spines receive one input spike as stimulation. The coefficients $\gamma_p$ and $\gamma_d$ correspond to the strength of potentiation and depression, respectively. Distance between spines is 1 μm. Letters P and D denote potentiation and depression, respectively. Combinations of these letters indicate the order of occurrence within the total time window from −100 to 100 ms (for calcium-dependent threshold, see Supplementary Figs. 1 and 3). Note that spine 1 is on the left and spine 2 is on the right side of the dendritic branch, and the timing is referenced to the input time of spine 1. **B** Patterns of calcium-based input timing synaptic plasticity for different inter-spine distance between spine 1 and spine 2 (1,2, and 3 μm). Due to the symmetry between spines 1 and 2, only the synaptic weight of spine 1 is shown. **C** Triplet-dependent heterosynaptic plasticity. Emergent plasticity patterns after stimulation of two spines at various time differences, with one spine receiving two input spikes and the second spine receiving one spike with a delay. The inter-input interval is 20 ms (for other calcium-dependent parameters, see Supplementary Figs. 5 and 6).

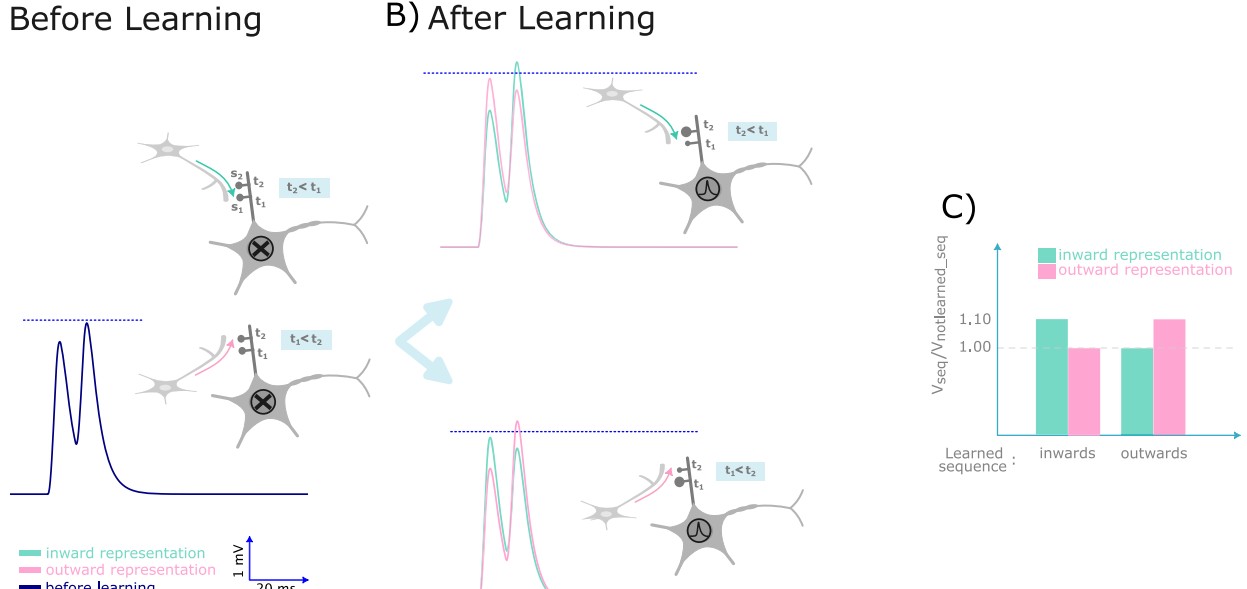

**Fig. 5 | Somatic voltage during sequence selectivity via heterosynaptic plasticity.** **A** The membrane potential of the soma before learning. The blue line shows the somatic response in the absence of calcium-dependent learning, resulting in no somatic spike. The dashed line shows the spiking threshold of the soma. **B** Somatic membrane potential after learning of the inward sequence. The green curve shows the response when the inward signal is presented to the dendrite after learning, while the pink curve shows the response when the outward signal is presented. The former leads to somatic firing. Similarly, the somatic response following learning of the outward sequence is shown in the bottom panel. In this case, the neuron, which has learned the outward signal, fires only when the outward sequence is presented to the dendrite. Note that the neuron's response can be modified by changing the location of its spines or the timing of the presynaptic partner. The blue dash-line represents the threshold for somatic firing. **C** The ratio of the somatic response to the learned sequence versus the non-learned sequence. The left bars shows the ratios for the protocol in which the inward sequence was presented during the learning phase, while the right bars illustrates the ratios when the outward pattern was presented during the learning phase.

potentiation (sLTP) using glutamate uncaging, showing that the number and arrangement of stimulated spines influence plasticity outcomes. Their mathematical model incorporated two types of proteins, but focused on abstract representations of signaling processes and potentiation, without reproducing synaptic depression within stimulated clusters. In contrast, our framework successfully replicated this phenomenon, which was experimentally observed by Oh et al.[38]. A recent study by Tsimring et al.[60] monitored structural and functional turnover of dendritic spines during the critical period, finding that spine retention is strongly dependent on calcium activity and highlighting the crucial role of heterosynaptic plasticity[60]. However, their experimental design may have led to the misclassification of spines[60], and their focus on functional role and pre- and postsynaptic elements makes direct comparison with our model challenging.

While our objective was to develop a computational model that balances biological realism with computational practicability, it is essential to recognize the possibility of further improvements. Future improvements could incorporate more intricate and detailed calcium sources, such as calcium-dependent channels or organelles that play a role in synaptic and dendritic calcium dynamics. For example, the rise in cytosolic $Ca^{2+}$ concentration is driven by the influx of extracellular $Ca^{2+}$ through open membrane channels within synapses, but could also be modulated by calcium channels along the dendritic cell membrane[35]. Intracellular $Ca^{2+}$ release from stores such as the endoplasmic reticulum (ER), mitochondria, and acidic $Ca^{2+}$ stores (e.g., lysosomes and endosomes) can also vary intracellular $Ca^{2+}$ concentration and signaling[35,36]. Ryanodine receptors (RyRs) are also pivotal $Ca^{2+}$ release channels within the ER, contributing to both NMDAR-dependent LTD and LTP under low-frequency stimulations[35,61]. RyRs are activated by $Ca^{2+}$ influx and can amplify the effect of incoming $Ca^{2+}$ signals, potentially lowering the threshold for LTP in neighboring synapses[36,61]. Calcium signaling can also occur through store-operated $Ca^{2+}$ entry (SOCE), a mechanism independent of neuronal activity, and facilitates $Ca^{2+}$ influx from the extracellular space. Both RyR-

mediated $Ca^{2+}$ release and SOCE play significant roles in shaping synaptic plasticity[36,61]. Furthermore, pre- and postsynaptic $Ca^{2+}$ stores are subject to dynamic regulation, influencing the expression of synaptic plasticity[36]. Given more detailed knowledge about the positioning, functioning, and dynamics of such $Ca^{2+}$ stores, our model can be extended to obtain a more complete picture of the functional implications of calcium signals.

The dendritic and synaptic morphology also has a large impact on observed plasticity patterns. Similar to a study showing the influence of spine density on the diffusion of $Cl^-$ ions[62], spines can act as barriers for diffusion, reducing the effective diffusion constant, which directly influences the pattern of homo- and heterosynaptic plasticity. Furthermore, changes in the thickness of the spine neck after LTP induction[63] can modulate the flux of $Ca^{2+}$ between spine and dendrite, influencing how strongly a synapse can be affected by heterosynaptic plasticity. Similarly, the positioning of receptors or the spine apparatus can modulate the flux of $Ca^{2+}$ between spine and dendritic segment[64–66]. It is worth noting that aging can change spine morphology and, consequently, synaptic plasticity[67]. However, in our study, we focused on early developmental stages, during which calcium is less confined to the spine head[26,63]. One limitation of our model is that it does not include the influence of back-propagating action potentials (bAPs) in the learning rule. Wright et al.[68] propose two distinct plasticity mechanisms in apical and basal dendrites, differing in their dependence on local co-activity and postsynaptic action potentials. However, their study found no evidence of precise millisecond-scale timing differences between somatic spikes and synaptic inputs in either apical or basal dendrites. This finding is consistent with previous research showing that bAPs are not universally present across neuron types and dendritic areas, likely due to variations in voltage-gated ion channels, such as $Na^+$ and $Ca^{2+}$[69]. Moreover, it has been established that bAPs are not a prerequisite for LTP[70], highlighting the complexity of synaptic plasticity mechanisms. To investigate this complex interplay between plasticity and bAPs, our model could be extended by such mechanisms using established mathematical descriptions of bAPs[71,72] or dendritic spikes[48,73].

We demonstrated the influence of input location and spine arrangement on emergent plasticity patterns in Supplementary Fig. 7. Our study focused on excitatory synapses, which predominantly host spines[21]. However, Agnes and Vogels[74] emphasized the importance of balancing excitatory and inhibitory synapses for stable synaptic weight profiles. Our model can be improved by incorporating distance-dependent spatial interaction strengths for proximal and distal dendrites, addressing a limitation of the abstract compartmental model used by Agnes and Vogels[74]. Tsimring et al.[60] also found that calcium activity plays a crucial role in shaping spine development, leading to synchronized spine activity during maturation, although with often imperfectly matched orientation and direction selectivity. Moreover, their study examined the effect of somatic output on calcium activity and spine properties via backpropagation signals[60], a process that is not accounted for in our current model. Somashekar and Bhalla[75] demonstrated that a bistable switch in dendritic branches can selectively respond to ordered sequences, but their study lacked a clear learning mechanism to explain this phenomenon and associated synaptic weight changes. Predictive error computation is attributed to discrepancies between top-down and bottom-up signals, with basal and apical dendrites receiving different inputs[76,77]. In general, extending our model by such discussed principles can provide a more detailed understanding about the role of dendrites in diverse types of neuronal computation.

Taken together, our results indicate that the intricate, nonlinear nature of the spatial and temporal integration of molecular signals introduces a level of intricacy that can produce elaborate patterns of synaptic plasticity. Especially, the sensitivity to input spike timing equips neuronal systems with a large and rich repertoire of learning rules, whose computational potential has yet to be discovered. In a series of studies, one specific realization of such a heterosynaptic input-STDP rule[78] provides a glimpse into the computational benefits and potential technological applications of heterosynaptic learning mechanisms, such as active noise reduction[79], control of walking robots[80], or learning with optical fibers[81].

## Methods
### Model
We introduce a biophysical computational model of the dynamics of the concentration of free $Ca^{2+}$ ions in the dendritic shaft and dendritic spines, and resulting in synaptic weight changes (Fig. 1B). We model the dendrite as a cylindrical structure with radius $r_d$ that we divide into many small segments, each of length $dx$. For different setups, varying numbers of "average" spines are placed at selected dendritic segments. Assuming a spine is present at segment $x$, the diffusion equation governing the change of the concentration of free calcium within the dendritic shaft is modeled by the following equation:

$$\frac{\partial C_d(x, t)}{\partial t} = D \frac{\partial^2 C_d(x, t)}{\partial x^2} - J_{ds}(x, t) + J_{sd}(x, t) - \frac{C_d(x, t)}{\tau_{decay}}. \quad (2)$$

$C_d(x, t)$ represents the calcium concentration (in μM) at the specified time $t$ and segment $x$. The first term on the right-hand side describes the spatial diffusion of the dendritic calcium concentration. The last term on the right-hand side summarizes the loss of freely diffusing calcium due to binding to proteins or efflux into the extracellular space with time constant $\tau_{decay}$. If segment $x$ does not host a spine, the terms $J_{ds}$ and $J_{sd}$ are set to zero. Otherwise, calcium flux from the dendritic segment to the spine is defined by $J_{ds}(x, t)$, and $J_{sd}(x, t)$ is the flux from the spine to the dendritic compartment, following:

$$J_{ds}(x, t) = k_d C_d(x, t), \quad (3)$$

$$J_{sd}(x, t) = k_s C_s(x, t). \quad (4)$$

## Table 1 | Used parameters

| Parameter | Symbol | Value | Source |
|---|---|---|---|
| Diffusion coefficient of free $Ca^{2+}$ | $D$ | 220 $\frac{\mu m^2}{s}$ | 82,85,86 |
| Time constant of calcium leakage | $\tau_{decay}$ | 0.08 s | 24 |
| Neck length | $l_n$ | 0.5 μm | 87 |
| Neck radius | $a_s$ | 0.1 μm | 87 |
| Spine radius | $r_s$ | 0.34 μm | 87 |
| Dendrite radius | $r_d$ | 1 μm | 88 |

Rates $k_d$ and $k_s$ are given by the inverse of the mean first passage time for influx[82,83] to the spine and to the dendrite, respectively:

$$k_d = \frac{1}{\tau} = \frac{4 D a_s}{V_d}, \quad (5)$$

$$k_s = \frac{1}{\tau'}, \text{ with } \tau' = \frac{V_s}{4 D a_s} + \frac{l_n^2}{2D} \quad (6)$$

With $V_d = \pi r_d^2 l_d$ and $V_s = \frac{4}{3} \pi r_s^3$ representing the volumes of the dendritic segment and spine, respectively. Here, $r_s$ and $r_d$ denote the radii of the spine and dendritic compartments, $l_d$ is the length of the dendritic compartment, and $l_n$ is the length of the spine neck (see Table 1).

The calcium concentration in the spine at dendritic segment $x$ is given by the following equation, with the same leak constant $\tau_{decay}$ as the dendrite:

$$\frac{\partial C_s(x, t)}{\partial t} = J_{syn}(x, t) + J_{ds}(x, t) - J_{sd}(x, t) - \frac{C_s(x, t)}{\tau_{decay}}. \quad (7)$$

In spines, the synaptic electrical current evoked by a presynaptic spike induces a calcium influx described by:

$$J_{syn}(x, t) = \frac{\gamma I_{ext}(x, t)}{z F V_s}, \quad (8)$$

where $I_{ext}(t)$ is the synaptic input current, $F$ is the Faraday constant, and $z$ is the calcium valence. $\gamma$ is the fraction of electrical current converted to calcium current, which is reported to be 0.11[82].

$$I_{ext}(t) = \sum_i I_0 e^{-(t-t_i)/\tau_d} \Theta(t - t_i) \quad (9)$$

with $I_0 = 0.1$ pA, $\tau_d = 1$ ms, and presynaptic spike times $t_i$.

Following the calcium hypothesis of (homo)synaptic plasticity[12,14], we consider that a high calcium concentration $C_s$ in the spine, which can be caused by high input rates, induces potentiation and strengthens the synaptic weight, while a medium level of calcium induces depression and weakens synaptic strength. Following Graupner and Brunel[14], we use a threshold-based plasticity model in which the weight of the synapse or spine at segment $x$ is updated based on the level of its calcium concentration $C_s$:

$$\frac{dw(x, t)}{dt} = (1 - w(x, t)) \gamma_p \Theta[C_s(x, t) - \theta_p] - w(x, t) \gamma_d \Theta[C_s(x, t) - \theta_d]. \quad (10)$$

Here, $\theta_p$ and $\theta_d$ are thresholds for potentiation and depression, $\Theta$ is the Heaviside function, $\gamma_p$ and $\gamma_d$ are LTP and LTD constants. When the calcium concentration exceeds $\theta_p$ or $\theta_d$, potentiation or depression occurs, respectively (Fig. 1A).

### Statistics and reproducibility
To account for different dendritic spine arrangements, the simulation results shown in Fig. 4 were obtained by considering 20 different

configurations of spine placement. Each data point corresponds to one configuration. Synaptic weights were calculated as the average across these configurations, and the corresponding standard deviations are plotted. For the comparison between the simulation results and the reported experimental data, only approximate values extracted from figures in the original publications were used and cited in the main text. Because we did not have access to the original experimental datasets, this comparison is qualitative. All other data were generated through simulations and are available in an open repository[84]. The data can be obtained via the provided simulation code (see code availability). Other plots are based on simulations of two spines at fixed locations; therefore, no statistical tests were performed.

Numerical simulations were conducted by discretizing the diffusion partial differential equation temporally and spatially, using an implicit (backward) Euler scheme to ensure numerical stability and convergence (see Supplementary Methods for details).

### Reporting summary
Further information on research design is available in the Nature Portfolio Reporting Summary linked to this article.

### Data availability
All data generated or analyzed during this study can be found at https://doi.org/10.5281/zenodo.18432747. Numerical Source data underlying all graphs in the manuscript can be found in the Supplementary data file.

### Code availability
The computer code underlying this study can be found in https://doi.org/10.5281/zenodo.18432747[84]. The simulations can be performed using Python 3.8 or 3.12.

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

## Acknowledgements
This work was supported by the German Research Foundation (Deutsche Forschungsgemeinschaft, DFG) through grants SFB1286/C01& Z01, TE 1172/7-1, as well as by the European Commission H2020 grant no. 899265 (ADOPD) and 945539 (HBP SGA3). We thank Arash Golmohammadi Naghshe for his valuable feedback and insightful comments.

## Author contributions
Conceptualization: C.T., S.Sc, and S.Sh; formal analysis: S.Sh, C.T.; funding acquisition: C.T.; investigation: C.T., S.Sh, and S.Sc; methodology: C.T., S.Sh, and S.Sc; project administration: C.T.; resources: C.T.; coding: S.Sh, S.Sc (with larger contribution by S.Sh); supervision: C.T.; visualization: S.Sh, C.T.; writing–original draft: S.Sh; writing–review and editing: C.T., S.Sh.

## Funding

## Competing interests
The authors declare no competing interests.

## Additional information
**Supplementary information** The online version contains Supplementary material available at https://doi.org/10.1038/s42003-026-09719-3.

