## [Transparent Peer Review File · Communications Biology]

Dendritic Heterosynaptic Plasticity Arises from Calcium-Based Input Learning

Corresponding Author: Ms shirin shafiee kamalabad

Version 0:

Reviewer comments:

Reviewer #1

(Remarks to the Author)

The manuscript by Shafiee et al. studies a generalization of the calcium-based synaptic plasticity model to a scenario where calcium can diffuse across the dendrite, and to/from the spine. In the model, this leads to a variety of non-additive effects where inputs to one synapse cause or modulate synaptic plasticity in neighboring synapses. The model is set up elegantly and investigated in interesting scenarios where the results of numerical simulations can be qualitatively related to experimental observations. This manuscript would be relevant to a broad neuroscience readership which includes both experimentalists and theoreticians. However, I have some concerns which I believe should be addressed prior to publication. Adequately addressing my concerns below could substantially improve the paper and make it a notable contribution to the literature in this field.

Major.

1. Model calibration and the effect of single spikes.

I commend the authors for obtaining a quantitative estimate of biophysically meaningful parameters (e.g., calcium diffusion constant, D) by matching the plasticity outcomes in the model to experimental observations (Fig. 3). However, these results rely crucially on model parameters that control the plasticity resulting from a given intracellular calcium concentration ($\theta_{d/p}$, $\gamma_{d/p}$), which were not sufficiently explored. It seems that the choices of those parameters might be quite far from physiologically meaningful values. For example, Fig. 1A shows that a single presynaptic spike leads to ~2% potentiation of the synapse, and ~2% depression of a neighboring synapse. This means that even a neuron with a low "baseline" firing rate of 1Hz would completely erase any memory stored by synapses downstream from it, and all their neighbors, within ~1 minute. This appears to be inconsistent with longitudinal observations of synapses, e.g., [Attardo et al. (2015) Nature] who showed a decorrelation timescale of a few days [see also Ziv et al (2013) Nature Neuroscience]. Reducing the rates $\gamma_{d/p}$ or increasing the thresholds $\theta_{d/p}$ may not be an easy fix. The reason is that, as the authors mention, the model as a whole is nonlinear so it is not obvious that the dependence on the parameters controlling diffusion would remain the same if the parameters controlling plasticity changed substantially.

2. Nonlinear calcium transients.

An alternative approach for stabilizing synapses within the context of calcium-based plasticity models (which is also consistent with experiments) is to model the calcium dynamics themselves as a nonlinear process. In scenarios where the postsynaptic neuron does generate spikes, such a model was suggested by [Graupner, Brunel (2012) PNAS], quantitatively fit to data in [Inglebert et al. (2020) PNAS], and analyzed in a recurrent network in [Wang, Aljadeff (2022) PRL]. The biological mechanism mediating the nonlinearity in these studies is typically thought to be the BPAP. However, another mechanism which could underlie this nonlinearity are dendritic NMDA spikes [see e.g. Brandalise et al. (2016) Nature Communications], which can be generated independently of postsynaptic activity. The authors may wish to investigate a nonlinear version of their model, for example by adding a nonlinearity in Eq. (7). This approach could be especially useful if it is difficult to reconcile a model where

- A. plasticity is "slowed down" (decreasing $\gamma_{d/p}$, increasing $\theta_{d/p}$) so that memories do not get erased immediately by baseline activity,
- B. calcium diffusion dynamics give realistic parameters (like in the current version).

3. Unified somatic/dendritic model.

The authors have chosen to focus only on homo and hetero synaptic plasticity arising independently of the postsynaptic neural activity. A detailed investigation of interactions between those forms of plasticity and plasticity rules that do depend on postsynaptic spikes may be beyond the scope of the current paper. Nevertheless I believe that proposing what an extension of the model that does take into account postsynaptic activity would substantially strengthen the work. Some progress in that direction already appears in the manuscript [Eq. (10)]. A recent paper [Wright et al. (2025) Science] has identified distinct plasticity rules which operate in different regions of the dendritic tree (i.e., closer or further away from the soma). If the authors can propose an extension of their model, their work would be much more relevant to modelers and experimentalists alike that are interested in investigating the combined effect of distinct plasticity rules in neural circuits.

Minor.

4. The authors do a good job of discussing the relevance of their work to biological processes and mechanisms at the subcellular scale (e.g., ER and mitochondria buffering of calcium and its effect on plasticity). Their work would be better situated within the literature if they expanded the discussion in the last paragraph about the relationship between their work and functional properties of circuits that were studied at the level of dendrites -- for example,

- A. Heterogeneous distribution of orientation selectivity of nearby synapses onto single neurons (related to Fig. 3)
- B. Involvement of dendritic compartments in prediction error computations
- C. Sequence generation (related to Fig. 5)

5. "different levels of input frequency"  "different input frequencies"

6. "are set equals zero"  "are set to zero" / "are equal to zeros"

7. "We ensure that at least three non-stimulated spines are situated outside the group of stimulated spines on each side" What motivates this choice of driving synapses with correlated inputs within a "compact" region? Is this based on a specific experimental paper? If not, what happens if this assumption is relaxed?

8. In Fig. 3B,C,D, it is not clear whether each point represents a synapse of one type or average over synapses of the same type in a given simulation. If the latter, then the authors should additionally show the variability of individual synapses. The points themselves are very hard to see. Same comments for Fig. S7 and S8.

Reviewer #3

(Remarks to the Author)

Recommendation: revision

The authors present a model of calcium dynamics in stimulated and unstimulated spines, and a learning rule based on the dynamics of calcium concentration. They demonstrate that the model can reproduce a set of diverse experimental results; specifically, dependence of the outcome of plasticity on the pattern of stimulation.

The paper is interesting and important since the presented model is generic with respect to specific sources of calcium, which makes it a good starting point for extending it by implementing specific calcium channels and other sources and their combinations.

There are, however, several points which need to be clarified.

One general concern is that the model introduces excessive capability of synapses/spines for plasticity - with (almost?) any single spike leading to plasticity. What are than parameters for a stable regime? More specifically, what would be a set of parameters with which spines change after strong stimulation, but remain stable during background/ "working" stimulation?

In the discussion, please extend discussion of what your new model adds/shows, and how it relates to prion experimental and model studies. In the present form, discussion is mostly about possible future extensions of the model (which is good and useful, but can't be the main part of the discussion).

Presentation of results.

On several occasions, self-evident things are resented as results. E.g. p. 5

"If we reduce the effective diffusion constant such that heterosynaptic plasticity will have a negligible effect on the plasticity pattern, model results always show only homosynaptic potentiation and no plasticity at unstimulated spines" - this is by design property of the model;

" It is evident that increasing γ_d enhances the level of depression, ... There are some intermediate values of γ_d and γ_p at which the temporal windows are becoming more complex."

Again, these are per design properties.

What is important here, is that there is a range of γ_d and γ_p values in which plasticity windows show more complex pattern;

please elaborate on this.

Figures.

Fig. 3

Axes legends are hardly visible.

Please indicate source of experimental data also in the figure legend.

Fig 4.

Why plasticity is called heterosynaptic, while both spines were stimulated?

Axes legends are hardly visible.

In A, Y-scale might be "Synaptic weight change"

In A, expand Y-scale, e.g. 0.9 - 1.15; with the present scale most space in the plots is empty, and difference between curves is hard to see.

Also, think about presenting plots in a systematic order (it looks like time windows do change systematically with a shift along X or Y-axis).

In B, three lines are difficult to distinguish; consider adding color or line-style, or some other labels.

In the stimulation schemes (all figures), consider positioning spines vertically, to show stimulation on the common time-scale.

Fig. 5.

Please show somatic responses to inwards and outwards stimulation patterns before and after learning.

Also, discuss and explain these non-trivial results.

Finally, language needs improvement.

The use of lab jargon should be avoided. E.g. "input spikes at a spine" is kind of nonsense; presynaptic spikes as such do not reach the postsynapse (spines).

Version 1:

Reviewer comments:

Reviewer #3

(Remarks to the Author)

The authors did good job addressing concerns from previous review. I have no further comments.

In my opinion, the authors adequately addressed concerns of the reviewer 1 (I believe I did briefly look through them already when reading the revised paper and author's response to my comments; and some of my comments on the original submission were actually similar to those of reviewer 1, and they were addressed in the revised paper).

I am confirming that I can recommend the revised version of the paper for publication.

List of Changes

Calcium-based input timing learning

2025-09-08

Dear reviewers, dear editors,

We would like to thank you for the appreciation of our work as well as for your detailed comments, which were very helpful to improve the quality of our study.

We have substantially revised the manuscript, including reformulation of our claims and adding further analysis to support our results. Besides the issues pointed out by the reviewers, we have applied minor changes for overall improvement throughout the manuscript. All changes in the manuscript are highlighted in blue.

You can find our updated simulation code and analysis scripts here: <https://github.com/Shirin1993/Calcium-based-input-timing-learning>. Please note that the numbering of references in this list of changes is different to that in the manuscript.

We thank you for your consideration and hope that you will now find the manuscript in an adequate state for publication.

Best regards,
Shirin Shafiee and Christian Tetzlaff

Reviewer #1

Remarks to the Author:

The manuscript by Shafiee et al. studies a generalization of the calcium-based synaptic plasticity model to a scenario where calcium can diffuse across the dendrite, and to/from the spine. In the model, this leads to a variety of non-additive effects where inputs to one synapse cause or modulate synaptic plasticity in neighboring synapses. The model is set up elegantly and investigated in interesting scenarios where the results of numerical simulations can be qualitatively related to experimental observations. This manuscript would be relevant to a broad neuroscience readership which includes both experimentalists and theoreticians. However, I have some concerns which I believe should be addressed prior to publication. Adequately addressing my concerns below could substantially improve the paper and make it a notable contribution to the literature in this field.

Response:

We would like to thank the reviewer for the positive feedback and thoughtful comments. Please find in the following a detailed point-by-point response, along with corresponding revisions of the manuscript.

Reviewer:

Major comments:

1. Model calibration and the effect of single spikes.

I commend the authors for obtaining a quantitative estimate of biophysically meaningful parameters (e.g., calcium diffusion constant, D) by matching the plasticity outcomes in the model to experimental observations (Fig. 3). However, these results rely crucially on model parameters that control the plasticity resulting from a given intracellular calcium concentration ($\theta_{d/p}$, $\gamma_{d/p}$), which were not sufficiently explored. It seems that the choices of those parameters might be quite far from physiologically meaningful values. For example, Fig. 1A shows that a single presynaptic spike leads to 2% potentiation of the synapse, and 2% depression of a neighboring synapse. This means that even a neuron with a low "baseline" firing rate of 1Hz would completely erase any memory stored by synapses downstream from it, and all their neighbors, within 1 minute. This appears to be inconsistent with longitudinal observations of synapses, e.g., [Attardo et al. (2015) Nature] who showed a decorrelation timescale of a few days [see also Ziv et al (2013) Nature Neuroscience]. Reducing the rates $\gamma_{d/p}$ or increasing the thresholds $\theta_{d/p}$ may not be an easy fix. The reason is that, as the authors mention, the model as a whole is nonlinear so it is not obvious that the dependence on the parameters controlling diffusion would remain the same if the parameters controlling plasticity changed substantially.

2. Nonlinear calcium transients.

An alternative approach for stabilizing synapses within the context of calcium-based plasticity models (which is also consistent with experiments) is to model the calcium dynamics themselves as a nonlinear process. In scenarios where the postsynaptic neuron does generate spikes, such a model was suggested by [Graupner, Brunel (2012) PNAS], quantitatively fit to data in [Inglebert et al. (2020) PNAS], and analyzed in a recurrent network in [Wang, Aljadeff (2022) PRL]. The biological mechanism mediating the nonlinearity in these studies is typically thought to be the BPAP. However, another mechanism which could underlie this nonlinearity are dendritic NMDA spikes [see e.g. Brandalise et al. (2016) Nature Communications], which can be generated independently of postsynaptic activity. The authors may wish to investigate a nonlinear version of their model, for example by adding a nonlinearity in Eq. (7). This approach could be especially useful if it is difficult to reconcile a model where A. plasticity is "slowed down" (decreasing $\gamma_{d/p}$, increasing $\theta_{d/p}$) so that memories do not get erased immediately by baseline activity, B. calcium diffusion dynamics give realistic parameters (like in the current version).

Response:

We would like to thank you for your positive and constructive feedback on our work. In Comment 1 Reviewer #3 raised a similar point. Therefore, we provide here the same response.

First, we would like to clarify that the analyses in Fig. 3 were conducted using multi-spine stimulation, in line with experimental observations showing that significant synaptic weight changes typically require the activation of more than one spine. In contrast, to better visualize the influence of spike timing, we used lower potentiation and depression thresholds in Fig. 2. We have adapted the manuscript accordingly to make this point clearer in the discussion of Fig. 2.

In addition, following the reviewer's suggestion, we simulated single-spine stimulation using the same parameters ($\theta_{d/p}$, $\gamma_{d/p}$) calibrated in Fig. 3. Please see Fig. 2. Under these conditions, the weight of neighboring spines remained unchanged, indicating stable synaptic weights. At the stimulated spine, homosynaptic plasticity yield only modest changes of about 1%. These results confirm that in our setup

matching experimental results, single-spine stimulation does not result in unrealistic potentiation or depression patterns, consistent with experimental findings (Chater et al., 2022; Lee et al., 2016).

Figure 1. Synaptic weight changes after noisy inputs. Random stimulation of the 'Stim' group with 0.1 Hz results in no significant synaptic weight changes at stimulated and neighboring spines.

[Manuscript, pages 5, line 230-234:]

[...] "We initially stimulated the first spine (blue), but different to the previous protocol, here, this stimulation was followed by a stimulation of the second spine (orange) 15ms later. Compared to the first protocol, stimulation of the second spine completely changed the pattern of plasticity, triggering synaptic depression at spine 1 and potentiation of spine 2 (Fig. 2C, right). To better illustrate the impact of spike timing on synaptic plasticity, we intentionally adjusted the potentiation and depression thresholds in Fig. 2 to be lower than those in Fig. 3, thereby amplifying the visibility of subtle timing-dependent changes."

Reviewer:

3. Unified somatic/dendritic model. *The authors have chosen to focus only on homo and hetero synaptic plasticity arising independently of the postsynaptic neural activity. A detailed investigation of interactions between those forms of plasticity and plasticity rules that do depend on postsynaptic spikes may be beyond the scope of the current paper. Nevertheless I believe that proposing what an extension of the model that does take into account postsynaptic activity would substantially strengthen the work. Some progress in that direction already appears in the manuscript [Eq. (10)]. A recent paper [Wright et al. (2025) Science] has identified distinct plasticity rules which operate in different regions of the dendritic tree (i.e., closer or further away from the soma). If the authors can propose an extension of their model, their work would be much more relevant to modelers and experimentalists alike that are interested in investigating the combined effect of distinct plasticity rules in neural circuits.*

Response:

Thank you for pointing this out. We extended our discussion section to discuss potential extensions and influences of postsynaptic activity events like back-propagating action potentials.

[Manuscript, page 9-10, line 502-521:]

"Furthermore, changes in the thickness of the spine neck after LTP induction (Kruijssen and Wierenga, 2019) can modulate the flux of Ca^{2+} between spine and dendrite, influencing how strongly a synapse can be affected by heterosynaptic plasticity. Similarly, the positioning of receptors or the spine apparatus can modulate the flux of Ca^{2+} between spine and dendritic segment (Breit et al., 2018; Noguchi et al., 2005; Rosado et al., 2022). It is worth noting that aging can change spine morphology and, consequently, synaptic plasticity (Bloss et al., 2011). However, in our study, we focused on early developmental stages, during which calcium is less confined to the spine head (Kruijssen and Wierenga, 2019; Lee et al., 2016).

One limitation of our model is that it does not include the influence of back-propagating action potentials (bAPs) in the learning rule. Wright et al., 2025 propose two distinct plasticity mechanisms in apical and basal dendrites, differing in their dependence on local co-activity and postsynaptic action potentials. However, their study found no evidence of precise millisecond-scale timing differences between somatic spikes and synaptic inputs in either apical or basal

dendrites. This finding is consistent with previous research showing that bAPs are not universally present across neuron types and dendritic areas, likely due to variations in voltage-gated ion channels such as Na^+ and Ca^{2+} (Waters et al., 2005). Moreover, it has been established that bAPs are not a prerequisite for long-term potentiation (Golding et al., 2002), highlighting the complexity of synaptic plasticity mechanisms. To investigate this complex interplay between plasticity and bAPs, our model could be extended by such mechanisms using established mathematical descriptions of bAPs (Kornijcuk et al., 2020; Rackham et al., 2010) or dendritic spikes (Bono and Clopath, 2017; Legenstein and Maass, 2011).

Minor

Reviewer:

4. *The authors do a good job of discussing the relevance of their work to biological processes and mechanisms at the subcellular scale (e.g., ER and mitochondria buffering of calcium and its effect on plasticity). Their work would be better situated within the literature if they expanded the discussion in the last paragraph about the relationship between their work and functional properties of circuits that were studied at the level of dendrites – for example,*
A. *Heterogeneous distribution of orientation selectivity of nearby synapses onto single neurons (related to Fig. 3)*
B. *Involvement of dendritic compartments in prediction error computations*
C. *Sequence generation (related to Fig. 5)*

Response:

We would like to thank you for your valuable input. We have extended the Discussion section accordingly, incorporating the reviewer’s suggestions as follows.

[Manuscript, page 10, line 522-549:]

”We demonstrated the influence of input location and spine arrangement on emergent plasticity patterns in Fig. S7. Our study focused on excitatory synapses, which predominantly host spines (Yuste, 2023). However, Agnes and Vogels, 2024 emphasized the importance of balancing excitatory and inhibitory synapses for stable synaptic weight profiles. Our model can be improved by incorporating distance-dependent spatial interaction strengths for proximal and distal dendrites, addressing a limitation of the abstract compartmental model used by Agnes and Vogels, 2024. Tsimring et al., 2025 also found that calcium activity plays a crucial role in shaping spine development, leading to synchronized spine activity during maturation, although with often imperfectly matched orientation and direction selectivity. Moreover, their study examined the effect of somatic output on calcium activity and spine properties via backpropagation signals Tsimring et al., 2025, a process that is not accounted for in our current model. Somashekar and Bhalla, 2025 demonstrated that a bistable switch in dendritic branches can selectively respond to ordered sequences, but their study lacked a clear learning mechanism to explain this phenomenon and associated synaptic weight changes. Predictive error computation is attributed to discrepancies between top-down and bottom-up signals, with basal and apical dendrites receiving different inputs (Keller and Mrsic-Flogel, 2018, Guerguiev et al., 2017). In general, extending our model by such discussed principles can provide a more detailed understanding about the role of dendrites in diverse types of neuronal computation.”

Reviewer:

5. *”different levels of input frequency” –> ”different input frequencies”*

Response:

[Manuscript, page 2, line 73:] Thank you. We adapted the text accordingly.

Reviewer:

6. *”are set equals zero”  ”are set to zero” / ”are equal to zeros”*

Response:

[Manuscript, page 3, line 155:] Thank you. We adapted the main text accordingly.

Reviewer:

7. "We ensure that at least three non-stimulated spines are situated outside the group of stimulated spines on each side" What motivates this choice of driving synapses with correlated inputs within a "compact" region? Is this based on a specific experimental paper? If not, what happens if this assumption is relaxed?

Response:

This assumption is based on the experimental setup of Chater et al., 2022. Since we did not have access to the experimental data for the exact positions of spines, we arranged spines similar to their experimental images. Moreover, to account for variability of positioning of spines, we varied the distribution of the spines within this region. Furthermore, the spine density has been reported with a quite wide range from 1 to 3.1 spine / μm with the average of 2.0 spines / μm in CA1 pyramidal cells in rat hippocampus (Harris and Stevens, 1989) or about 20 spines / $10\mu\text{m}$ in layer II pyramidal cells in the mouse cortex (Ballesteros-Yáñez et al., 2006, but also see Megias et al., 2001). Following these experimental results, in our simulation, we placed 10 spines (7 stimulated and 3 unstimulated among them) in a 7 μm section of a branch, which gives us ~ 1.4 spines/ μm . We adapted the main text to clarify this point in the manuscript. [Manuscript, page 5, line 248-259:]

"Next, we extended our model considering a dendrite with multiple spines to match our model results to various experimental data sets (Chater et al. 2022; Oh et al. 2015; Tong et al. 2021). Note that the experimentally reported plasticity patterns after stimulation show a high degree of variety, lacking a coherent explanation. To integrate all three data sets, we modeled a 80 μm long dendrite hosting 16 randomly distributed spines that are governed by calcium-dependent synaptic plasticity (Fig. 3A). All protocols have been repeated for 20 different, random spine configurations, in which initial synaptic weights are randomly chosen representing different initial sizes and volumes of the spines in experiment. A random subset of 7 spines receives stimulation and forms in the following the group of stimulated spines (Stim, teal color). We ensure that at least three non-stimulated spines are situated outside the group of stimulated spines on each side ($U_{n_{out}}$, pastel yellow). All further spines are also non-stimulated and form the group of inner neighboring spines ($U_{n_{in}}$, coral pink). Note that the spatial distribution of spines is arranged such that a total of 10 spines, comprising 7 stimulated and 3 unstimulated ones, are confined to a 7 μm section of a dendritic branch, yielding a spine density of approximately 1.4 spines/ μm , following reported experimental values (Ballesteros-Yáñez et al., 2006; Chater et al., 2022; Harris and Stevens, 1989; Megias et al., 2001)."

Reviewer:

8. In Fig. 3B,C,D, it is not clear whether each point represents a synapse of one type or average over synapses of the same type in a given simulation. If the latter, then the authors should additionally show the variability of individual synapses. The points themselves are very hard to see. Same comments for Fig. S7 and S8.

Response:

[Manuscript, page 6:] In Fig. 3B,C and D, each point corresponds to the result of a simulation at that frequency for a specific arrangement of spines. Thus, for arrangement A and B of spines, the change of synaptic weights are calculated separately for each spine group (namely Stim, $U_{n_{in}}$ and $U_{n_{out}}$) and are shown as a point in each plot. The point size has been increased as suggested, and the corresponding captions have been adjusted accordingly in both figures.

Updated figure 3. Simulation and experimental results at different frequencies (2, 100, and 150 Hz). **A)** Arrangement of stimulated (in teal color) nearby unstimulated (in coral pink) and distant unstimulated spines (in pastel yellow). **B)-D)** Simulation results with low ($D=1 \frac{\mu m^2}{s}$) and physiological diffusion constant ($D=220 \frac{\mu m^2}{s}$) as well as experimental results. **B)** Simulation results with low input frequency as well as experimental observation (see Fig. 1 in Oh et al., 2013), **C)** simulation results at medium frequency and experimental observation (see Fig. 2. ciii in Tong et al., 2021), and **D)** simulation at high frequency as well as experimental observation (see Fig. S1 in Chater et al., 2022). Each data point corresponds to the synaptic weight change of a specific synapse type (Stim, Un_in, or Un_out) obtained after simulation at a particular frequency and with a specific spatial arrangement of spines. The standard deviation reflects the variability in synaptic weight changes obtained across different spatial arrangements of spines. **E)** Errors between simulation and experiments shown in B-D) for different diffusion constants (see synaptic weight patterns in Fig. S8). Black data points represent the total error for a specific spine arrangement across all frequencies and spine groups. Note that diffusion constants are plotted in logarithmic scale. The standard deviation represents the variability in error values obtained from different spatial arrangements of spines.

[Supplementary, page :]

Updated figure S7. Simulation results at in different diffusion constants and different frequencies (2, 100, and 150 Hz). Simulation results with low ($D=1$) and physiological diffusion constant ($D=220$). Simulation results with low input frequency (first column), medium frequency (second column), and high frequency (third column). Each data point corresponds to the synaptic weight change of a specific synapse type (Stim, Un_in, or Un_out) obtained after simulation at a particular frequency and with a specific spatial arrangement of spines. The standard deviation reflects the variability in synaptic weight changes obtained across different spatial arrangements of spines.

Reviewer #3

Reviewer:

Recommendation: revision

The authors present a model of calcium dynamics in stimulated and unstimulated spines, and a learning rule based on the dynamics of calcium concentration. They demonstrate that the model can reproduce a set of diverse experimental results; specifically, dependence of the outcome of plasticity on the pattern of stimulation.

The paper is interesting and important since the presented model is generic with respect to specific sources of calcium, which makes it a good starting point for extending it by implementing specific calcium channels and other sources and their combinations.

Response:

We would like to thank the reviewer for the positive feedback and valuable comments.

Reviewer:

1) One general concern is that the model introduces excessive capability of synapses/spines for plasticity - with (almost?) any single spike leading to plasticity. What are than parameters for a stable regime? More specifically, what would be a set of parameters with which spines change after strong stimulation, but remain stable during background/ "working" stimulation?

Response:

Thank you for point this out. In comments 1 and 2 Reviewer #1 raised a similar point. Therefore, we provide in the following the same response:

First, we would like to clarify that the analyses in Fig. 3 were conducted using multi-spine stimulation, in line with experimental observations showing that significant synaptic weight changes typically require the activation of more than one spine. In contrast, to better visualize the influence of spike timing, we used lower potentiation and depression thresholds in Fig. 2. We have adapted the manuscript accordingly to make this point clearer in the discussion of Fig. 2.

In addition, following the reviewer's suggestion, we simulated single-spine stimulation using the same parameters ($\theta_{d/p}$, $\gamma_{d/p}$) calibrated in Fig. 3. Please see Fig. 2. Under these conditions, the weight of neighboring spines remained unchanged, indicating stable synaptic weights. At the stimulated spine, homosynaptic plasticity yield only modest changes of about 1%. These results confirm that in our setup matching experimental results, single-spine stimulation does not result in unrealistic potentiation or depression patterns, consistent with experimental findings (Chater et al., 2022; Lee et al., 2016).

Figure 2. Synaptic weight changes after noisy inputs. Random stimulation of the 'Stim' group with 0.1 Hz results in no significant synaptic weight changes at stimulated and neighboring spines.

[Manuscript, pages 3-5:]

[...] "We initially stimulated the first spine (blue), but different to the previous protocol, here, this stimulation is followed by a stimulation of the second spine (orange) 15ms later. Compared to the first protocol, stimulation of the second spine completely changes the pattern of plasticity, triggering synaptic depression at spine 1 and potentiation of spine 2 (Fig. 2C, right). To better illustrate the impact of spike timing on synaptic plasticity, we intentionally adjusted the potentiation and depression thresholds in Fig. 2 to be lower than those in Fig. 3, thereby amplifying the visibility of subtle timing-dependent changes."

Reviewer:

2) *In the discussion, please extend discussion of what your new model adds/shows, and how it relates to prion experimental and model studies. In the present form, discussion is mostly about possible future extensions of the model (which is good and useful, but can't be the main part of the discussion).*

Response:

Thank you for the suggestion. We have extended the discussion section to include a more detailed examination of previous experimental and computational studies, and provided a comparative discussion with our current model.

[Manuscript, page 8, line 427-448:]

"We developed a computational model of molecular diffusion dynamics in a piece of dendrite and connected spines. Using a diffusion constant similar to that of calcium, our model can match results from several experimental studies of homo- and heterosynaptic plasticity. Our model also demonstrates that, given the sensitivity of calcium influx to the timing of presynaptic input spikes, the triggered homo- and heterosynaptic plasticity can lead to complex input-spike-timing-dependent plasticity patterns. This provides a rich repertoire of calcium-based learning rules without the need for postsynaptic spiking.

Previous experimental studies have reported seemingly contradictory results (Chater and Goda, 2021; Oh et al., 2015; Tong et al., 2021), yet a comprehensive mechanism explaining these discrepancies has not been proposed. Chater et al., 2022 investigated structural long-term potentiation (sLTP) using glutamate uncaging, showing that the number and arrangement of stimulated spines influence plasticity outcomes. Their mathematical model incorporated two types of proteins, but focused on abstract representations of signaling processes and potentiation, without reproducing synaptic depression within stimulated clusters (Chater et al., 2024). In contrast, our framework successfully replicated this phenomenon, which was experimentally observed by Oh et al., 2015. A recent study by Tsimring et al., 2025 monitored structural and functional turnover of dendritic spines during the critical period, finding that spine retention is strongly dependent on calcium activity and highlighting the crucial role of heterosynaptic plasticity (Tsimring et al., 2025). However, their experimental design may have led to the misclassification of spines (Tsimring et al., 2025), and their focus on functional role and pre- and postsynaptic elements makes direct comparison with our model challenging.

While our objective was to develop a computational model that balances biological realism with computational practicability, it is essential to recognize the possibility of further improvements. Future improvements could incorporate more intricate and detailed calcium sources, such as calcium-dependent channels or organelles that play a role in synaptic and dendritic calcium dynamics."

Reviewer:

3) *Presentation of results. On several occasions, self-evident things are resented as results. E.g. p. 5 "If we reduce the effective diffusion constant such that heterosynaptic plasticity will have a negligible effect on the plasticity pattern, model results always show only homosynaptic potentiation and no plasticity at unstimulated spines" - this is by design property of the model;*

"It is evident that increasing γ_d enhances the level of depression, ... There are some intermediate values of γ_d and γ_p at which the temporal windows are becoming more complex." Again, these are per design properties. What is important here, is that there is a range of γ_d and γ_p values in which plasticity windows show more complex pattern; please elaborate on this.

Response:

Thank you very much. We agree to your point and have adapted the text to better clarify our findings.

[Manuscript, page 5, line 303-311:]

"Furthermore, the induction of heterosynaptic plasticity is essential for all protocols. If we reduce the effective diffusion constant such that heterosynaptic plasticity will have a negligible

effect on the plasticity pattern, model results always show only homosynaptic potentiation and no plasticity at unstimulated spines (left, Fig. 3B-D).

We repeated all protocols with different values of the diffusion constant ($D = 1, 30, 65, 140, 220, 260, 300, 350, 400, 440 \frac{\mu\text{m}^2}{\text{s}}$) and 20 random configurations each (Fig. 3E). Then, we counted the number of instances where the resulting plasticity patterns matched the plasticity pattern of the corresponding experiment for each spine group (i.e., Stim, Un_{in}, and Un_{out}, as shown in the last column of Fig. 3B, C, and D), and derived a matching error between 0 and 1. It is evident from Fig. 3E that the lowest error value corresponds to a range of diffusion constants near the calcium diffusion constant. *Although reducing the diffusion constant in our model diminishes the heterosynaptic effect by design. However, it is worth noting that this change not only impairs heterosynaptic plasticity at the corresponding frequency, but also affects the homosynaptic pattern, which no longer matches the reported experimental results. For instance, in Fig. 3B (left panel), stimulated spines undergo potentiation at $D=1$, whereas at $D=220$ they undergo depression, consistent with experimental observations. This result supports the model assumption of calcium being one of the main candidates to communicate heterosynaptic plasticity.*

[Manuscript, page 5,7, line 331-346:]

Fig. 4A shows the categorized patterns of synaptic weights at different input time differences. It is evident that increasing γ_d enhances the level of depression, thereby shifting the temporal window towards depression for all time differences (green region in Fig. 4A). Similarly, increasing the potentiation coefficient, γ_p , shifts the temporal window towards potentiation for all time differences (red region in Fig. 4A). There are intermediate values of γ_d and γ_p at which the temporal windows become more complex. *It is noteworthy that the majority of existing experimental and theoretical studies have primarily focused on pair activity of neurons (Inglebert et al., 2020; Tazerart et al., 2020) and external calcium (Inglebert et al., 2020). Consequently, the parameter values corresponding to $\theta_{p/d}$ and $\gamma_{p/d}$ cannot be directly compared to these studies. Furthermore, previous work, such as Inglebert et al., 2020, which employed parameter fitting, still suggests multiple ranges of values for some of these parameters. Other computational studies (Graupner and Brunel, 2012; Hiratani and Fukai, 2017) have also focused on pairing protocols and proposed different values for these parameters. Although we explored a wide range of $\theta_{p/d}$ and $\gamma_{p/d}$ and selected values that match different experimental observations, a fair comparison and quantification of the parameters of our model necessitate a compatible experimental setup.*

Reviewer:

4) **Fig. 3** Axes legends are hardly visible. Please indicate source of experimental data also in the figure legend.

Response:

Thank you for pointing this out. We have changed the font size for better visibility. The citations are now declared in the caption of figures. Please see response to Comment 8, Reviewer 1 for further changes to the figure.

Updated figure 3. Simulation and experimental results at different frequencies (2, 100, and 150 Hz). **A)** Arrangement of stimulated (in teal color) nearby unstimulated (in coral pink) and distant unstimulated spines (in pastel yellow). **B)-D)** Simulation results with low ($D=1 \frac{\mu m^2}{s}$) and physiological diffusion constant ($D=220 \frac{\mu m^2}{s}$) as well as experimental results. **B)** Simulation results with low input frequency as well as experimental observation (see Fig. 1 in Oh et al., 2013), **C)** simulation results at medium frequency and experimental observation (see Fig. 2. ciii in Tong et al., 2021), and **D)** simulation at high frequency as well as experimental observation (see Fig. S1 in Chater et al., 2022). Each data point corresponds to the synaptic weight change of a specific synapse type (Stim, Un_in, or Un_out) obtained after simulation at a particular frequency and with a specific spatial arrangement of spines. The standard deviation reflects the variability in synaptic weight changes obtained across different spatial arrangements of spines. **E)** Errors between simulation and experiments shown in B-D) for different diffusion constants (see synaptic weight patterns in Fig. S8). Black data points represent the total error for a specific spine arrangement across all frequencies and spine groups. Note that diffusion constants are plotted in logarithmic scale. The standard deviation represents the variability in error values obtained from different spatial arrangements of spines.

Reviewer:

5) **Fig 4.** *Why plasticity is called heterosynaptic, while both spines were stimulated?*

Axes legends are hardly visible. In A, Y-scale might be "Synaptic weight change" In A, expand Y-scale, e.g. 0.9 - 1.15; with the present scale most space in the plots is empty, and difference between curves is hard to see. Also, think about presenting plots in a systematic order (it looks like time windows do change systematically with a shift along X or Y-axis).

In B, three lines are difficult to distinguish; consider adding color or line-style, or some other labels.

In the stimulation schemes (all figures), consider positioning spines vertically, to show stimulation on the common time-scale.

Response:

Although both spines are stimulated, in addition to the homosynaptic plasticity that has led to the weight change of the synapse, the heterosynaptic effect has modified the magnitude and/or direction of the weight change, depending on the inter-spine distance or time window between the activities of the spines. Therefore, we talk here about heterosynaptic plasticity.

Regarding the scale, we expanded the Y-scale for better readability. Since we had plotted the curves for all pairs of parameters, we chose the original Y-scale to facilitate comparison among all curves, as some values reached beyond 1.1 or below 0.9, respectively.

A)

B)

C)

Updated figure 4. Various curves of heterosynaptic, input-timing plasticity. **A)** Emergent temporal windows after stimulation of two spines at various time differences. Both spines receive one input spike as stimulation. The coefficients γ_p and γ_d correspond to the strength of potentiation and depression, respectively. Distance between spines is $1 \mu\text{m}$. Letters P and D denote potentiation and depression, respectively. Combinations of these letters indicate the order of occurrence within the total time window from -100 to 100 ms (For calcium-dependent threshold, see Fig. S1, S3). Note that spine 1 is on the left and spine 2 is on the right side of the dendritic branch, and the timing is referenced to the input time of spine 1. **B)** Patterns of calcium-based input timing synaptic plasticity for different inter-spine distance between spine 1 and spine 2 ($1, 2$ and $3 \mu\text{m}$). Due to the symmetry between spines 1 and 2, only the synaptic weight of spine 1 is shown. **C)** Triplet-dependent heterosynaptic plasticity. Emergent plasticity patterns after stimulation of two spines at various time differences with one spine receiving two input spikes and the second spine receiving one spike with a delay. The inter-input interval is 20 ms (For other calcium-dependent parameters, see Fig. S5, S6).

Reviewer:

6) **Fig. 5.** Please show somatic responses to inwards and outwards stimulation patterns before and after learning. Also, discuss and explain these non-trivial results.

Response:

We adapted the figure as suggested and discuss now the results in more detail. Consequently, Fig. 9 from the previous version of the supplementary material has been removed.

Updated figure 5. Somatic voltage during sequence selectivity via heterosynaptic plasticity.

A) The membrane potential of the soma before learning. The blue line shows the somatic response in the absence of calcium-dependent learning, resulting in no somatic spike. The dashed line shows the spiking threshold of the soma. **B) Top:** Somatic membrane potential after learning of the inward sequence. The green curve shows the response when the inward signal is presented to the dendrite after learning, while the pink curve shows the response when the outward signal is presented. The former leads to somatic firing. **Bottom:** Similar to the **top**, but for the response of the soma after learning of the outward sequence. In this case, the neuron, which has learned the outward signal, fires only when the outward sequence is presented to the dendrite. Note that the neuron's response can be modified by changing the location of its spines or the timing of the presynaptic partner. The blue dash-line represents the threshold for somatic firing. **C)** The ratio of the somatic response to the learned sequence versus the non-learned sequence. The left bars show the ratios for the protocol in which the inward sequence was presented during the learning phase, while the right bars illustrate the ratios when the outward pattern was presented during the learning phase.

Reviewer:

7) Finally, language needs improvement. The use of lab jargon should be avoided. E.g. "input spikes at a spine" is kind of nonsense; presynaptic spikes as such do not reach the postsynapse (spines).

Response:

Thank you. We carefully adapted the text. Please note that we did not highlight language changes in the main text.

References

- Agnes, E. J., & Vogels, T. P. (2024). Co-dependent excitatory and inhibitory plasticity accounts for quick, stable and long-lasting memories in biological networks. *Nature Neuroscience*, 27(5), 964–974.
- Ballesteros-Yáñez, I., Benavides-Piccione, R., Elston, G., Yuste, R., & DeFelipe, J. (2006). Density and morphology of dendritic spines in mouse neocortex. *Neuroscience*, 138(2), 403–409.

- Bloss, E. B., Janssen, W. G., Ohm, D. T., Yuk, F. J., Wadsworth, S., Saardi, K. M., McEwen, B. S., & Morrison, J. H. (2011). Evidence for reduced experience-dependent dendritic spine plasticity in the aging prefrontal cortex. *Journal of Neuroscience*, *31*(21), 7831–7839.
- Bono, J., & Clopath, C. (2017). Modeling somatic and dendritic spike mediated plasticity at the single neuron and network level. *Nature communications*, *8*(1), 706.
- Breit, M., Kessler, M., Stepniewski, M., Vlachos, A., & Queisser, G. (2018). Spine-to-dendrite calcium modeling discloses relevance for precise positioning of ryanodine receptor-containing spine endoplasmic reticulum. *Scientific reports*, *8*(1), 15624.
- Chater, T. E., Eggl, M., Goda, Y., & Tchumatchenko, T. (2022). A quantitative rule to explain multi-spine plasticity. *bioRxiv*. <https://doi.org/10.1101/2022.07.04.498706>
- Chater, T. E., Eggl, M. F., Goda, Y., & Tchumatchenko, T. (2024). Competitive processes shape multi-synapse plasticity along dendritic segments. *Nature Communications*, *15*(1), 7572.
- Chater, T. E., & Goda, Y. (2021). My neighbour hetero—deconstructing the mechanisms underlying heterosynaptic plasticity. *Current Opinion in Neurobiology*, *67*, 106–114.
- Golding, N. L., Staff, N. P., & Spruston, N. (2002). Dendritic spikes as a mechanism for cooperative long-term potentiation. *Nature*, *418*(6895), 326–331.
- Graupner, M., & Brunel, N. (2012). Calcium-based plasticity model explains sensitivity of synaptic changes to spike pattern, rate, and dendritic location (vol 109, pg 3991, 2012). *Proceedings of the National Academy of Sciences of the United States of America*, *109*(52), 21551–21551.
- Guerguiev, J., Lillicrap, T. P., & Richards, B. A. (2017). Towards deep learning with segregated dendrites. *elife*, *6*, e22901.
- Harris, K. M., & Stevens, J. K. (1989). Dendritic spines of ca 1 pyramidal cells in the rat hippocampus: Serial electron microscopy with reference to their biophysical characteristics. *Journal of Neuroscience*, *9*(8), 2982–2997.
- Hiratani, N., & Fukai, T. (2017). Detailed dendritic excitatory/inhibitory balance through heterosynaptic spike-timing-dependent plasticity. *Journal of Neuroscience*, *37*(50), 12106–12122.
- Inglebert, Y., Aljadeff, J., Brunel, N., & Debanne, D. (2020). Synaptic plasticity rules with physiological calcium levels. *Proceedings of the National Academy of Sciences*, *117*(52), 33639–33648.
- Keller, G. B., & Mrsic-Flogel, T. D. (2018). Predictive processing: A canonical cortical computation. *Neuron*, *100*(2), 424–435.
- Kornijcuk, V., Kim, D., Kim, G., & Jeong, D. S. (2020). Simplified calcium signaling cascade for synaptic plasticity. *Neural Networks*, *123*, 38–51.
- Kruijssen, D. L., & Wierenga, C. J. (2019). Single synapse ltp: A matter of context? *Frontiers in cellular neuroscience*, *13*, 496.
- Lee, K. F., Soares, C., Thivierge, J.-P., & Béique, J.-C. (2016). Correlated synaptic inputs drive dendritic calcium amplification and cooperative plasticity during clustered synapse development. *Neuron*, *89*(4), 784–799.
- Legenstein, R., & Maass, W. (2011). Branch-specific plasticity enables self-organization of nonlinear computation in single neurons. *Journal of Neuroscience*, *31*(30), 10787–10802.
- Megias, M., Emri, Z., Freund, T., & Gulyás, A. (2001). Total number and distribution of inhibitory and excitatory synapses on hippocampal ca1 pyramidal cells. *Neuroscience*, *102*(3), 527–540.
- Noguchi, J., Matsuzaki, M., Ellis-Davies, G. C., & Kasai, H. (2005). Spine-neck geometry determines nmda receptor-dependent ca²⁺ signaling in dendrites. *Neuron*, *46*(4), 609–622.
- Oh, W. C., Hill, T. C., & Zito, K. (2013). Synapse-specific and size-dependent mechanisms of spine structural plasticity accompanying synaptic weakening. *Proceedings of the National Academy of Sciences*, *110*(4), E305–E312.
- Oh, W. C., Parajuli, L. K., & Zito, K. (2015). Heterosynaptic structural plasticity on local dendritic segments of hippocampal ca1 neurons. *Cell reports*, *10*(2), 162–169.
- Rackham, O., Tsaneva-Atanasova, K., Ganesh, A., & Mellor, J. (2010). A ca²⁺-based computational model for nmda receptor-dependent synaptic plasticity at individual post-synaptic spines in the hippocampus. *Frontiers in synaptic neuroscience*, *2*, 1356.
- Rosado, J., Bui, V. D., Haas, C. A., Beck, J., Queisser, G., & Vlachos, A. (2022). Calcium modeling of spine apparatus-containing human dendritic spines demonstrates an “all-or-nothing” communication switch between the spine head and dendrite. *PLoS Computational Biology*, *18*(4), e1010069.
- Somashekar, B. P., & Bhalla, U. S. (2025). Discriminating neural ensemble patterns through dendritic computations in randomly connected feedforward networks. *eLife*, *13*, RP100664.
- Tazerart, S., Mitchell, D. E., Miranda-Rottmann, S., & Araya, R. (2020). A spike-timing-dependent plasticity rule for dendritic spines. *Nature communications*, *11*(1), 4276.

- Tong, R., Chater, T. E., Emptage, N. J., & Goda, Y. (2021). Heterosynaptic cross-talk of pre-and post-synaptic strengths along segments of dendrites. *Cell reports*, *34*(4).
- Tsimring, K., Jenks, K. R., Cusceddu, C., Heller, G. R., Ip, J. P. K., Gjorgjieva, J., & Sur, M. (2025). Large-scale synaptic dynamics drive the reconstruction of binocular circuits in mouse visual cortex. *Nature Communications*, *16*(1), 5810.
- Waters, J., Schaefer, A., & Sakmann, B. (2005). Backpropagating action potentials in neurones: Measurement, mechanisms and potential functions. *Progress in biophysics and molecular biology*, *87*(1), 145–170.
- Wright, W. J., Hedrick, N. G., & Komiyama, T. (2025). Distinct synaptic plasticity rules operate across dendritic compartments in vivo during learning. *Science*, *388*(6744), 322–328.
- Yuste, R. (2023). *Dendritic spines*. MIT press.

List of Changes

Calcium-based input timing learning

2026-01-11

Reviewer #1

Remarks to the Author:

The manuscript by Shafiee et al. studies a generalization of the calcium-based synaptic plasticity model to a scenario where calcium can diffuse across the dendrite, and to/from the spine. In the model, this leads to a variety of non-additive effects where inputs to one synapse cause or modulate synaptic plasticity in neighboring synapses. The model is set up elegantly and investigated in interesting scenarios where the results of numerical simulations can be qualitatively related to experimental observations. This manuscript would be relevant to a broad neuroscience readership which includes both experimentalists and theoreticians. However, I have some concerns which I believe should be addressed prior to publication. Adequately addressing my concerns below could substantially improve the paper and make it a notable contribution to the literature in this field.

Response:

We would like to thank the reviewer for the positive feedback and thoughtful comments. Please find in the following a detailed point-by-point response, along with corresponding revisions of the manuscript.

Reviewer:

Major comments:

1. Model calibration and the effect of single spikes.

I commend the authors for obtaining a quantitative estimate of biophysically meaningful parameters (e.g., calcium diffusion constant, D) by matching the plasticity outcomes in the model to experimental observations (Fig. 3). However, these results rely crucially on model parameters that control the plasticity resulting from a given intracellular calcium concentration ($\theta_{d/p}$, $\gamma_{d/p}$), which were not sufficiently explored. It seems that the choices of those parameters might be quite far from physiologically meaningful values. For example, Fig. 1A shows that a single presynaptic spike leads to 2% potentiation of the synapse, and 2% depression of a neighboring synapse. This means that even a neuron with a low "baseline" firing rate of 1Hz would completely erase any memory stored by synapses downstream from it, and all their neighbors, within 1 minute. This appears to be inconsistent with longitudinal observations of synapses, e.g., [Attardo et al. (2015) Nature] who showed a decorrelation timescale of a few days [see also Ziv et al (2013) Nature Neuroscience]. Reducing the rates $\gamma_{d/p}$ or increasing the thresholds $\theta_{d/p}$ may not be an easy fix. The reason is that, as the authors mention, the model as a whole is nonlinear so it is not obvious that the dependence on the parameters controlling diffusion would remain the same if the parameters controlling plasticity changed substantially.

2. Nonlinear calcium transients.

An alternative approach for stabilizing synapses within the context of calcium-based plasticity models (which is also consistent with experiments) is to model the calcium dynamics themselves as a nonlinear process. In scenarios where the postsynaptic neuron does generate spikes, such a model was suggested by [Graupner, Brunel (2012) PNAS], quantitatively fit to data in [Inglebert et al. (2020) PNAS], and analyzed in a recurrent network in [Wang, Aljadeff (2022) PRL]. The biological mechanism mediating the nonlinearity in these studies is typically thought to be the BPAP. However, another mechanism which could underlie this nonlinearity are dendritic NMDA spikes [see e.g. Brandalise et al. (2016) Nature Communications], which can be generated independently of postsynaptic activity. The authors may wish to investigate a nonlinear version of their model, for example by adding a nonlinearity in Eq. (7). This approach could be especially useful if it is difficult to reconcile a model where A. plasticity is "slowed down" (decreasing $\gamma_{d/p}$, increasing $\theta_{d/p}$) so that memories do not get erased immediately by baseline activity, B. calcium diffusion dynamics give realistic parameters (like in the current version).

Response:

We would like to thank you for your positive and constructive feedback on our work. In Comment 1 Reviewer #3 raised a similar point. Therefore, we provide here the same response.

First, we would like to clarify that the analyses in Fig. 3 were conducted using multi-spine stimulation, in line with experimental observations showing that significant synaptic weight changes typically require the activation of more than one spine. In contrast, to better visualize the influence of spike timing, we used lower potentiation and depression thresholds in Fig. 2. We have adapted the manuscript accordingly to make this point clearer in the discussion of Fig. 2.

In addition, following the reviewer’s suggestion, we simulated single-spine stimulation using the same parameters ($\theta_{d/p}$, $\gamma_{d/p}$) calibrated in Fig. 3. Please see Fig. 2. Under these conditions, the weight of neighboring spines remained unchanged, indicating stable synaptic weights. At the stimulated spine, homosynaptic plasticity yield only modest changes of about 1%. These results confirm that in our setup matching experimental results, single-spine stimulation does not result in unrealistic potentiation or depression patterns, consistent with experimental findings (Chater et al., 2022; Lee et al., 2016).

Figure 1. Synaptic weight changes after noisy inputs. Random stimulation of the 'Stim' group with 0.1 Hz results in no significant synaptic weight changes at stimulated and neighboring spines.

[Manuscript, pages 5, line 230-234:]

[...] ”We initially stimulated the first spine (blue), but different to the previous protocol, here, this stimulation was followed by a stimulation of the second spine (orange) 15ms later. Compared to the first protocol, stimulation of the second spine completely changed the pattern of plasticity, triggering synaptic depression at spine 1 and potentiation of spine 2 (Fig. 2C, right). To better illustrate the impact of spike timing on synaptic plasticity, we intentionally adjusted the potentiation and depression thresholds in Fig. 2 to be lower than those in Fig. 3, thereby amplifying the visibility of subtle timing-dependent changes.”

Reviewer:

3. Unified somatic/dendritic model. *The authors have chosen to focus only on homo and hetero synaptic plasticity arising independently of the postsynaptic neural activity. A detailed investigation of interactions between those forms of plasticity and plasticity rules that do depend on postsynaptic spikes may be beyond the scope of the current paper. Nevertheless I believe that proposing what an extension of the model that does take into account postsynaptic activity would substantially strengthen the work. Some progress in that direction already appears in the manuscript [Eq. (10)]. A recent paper [Wright et al. (2025) Science] has identified distinct plasticity rules which operate in different regions of the dendritic tree (i.e., closer or further away from the soma). If the authors can propose an extension of their model, their work would be much more relevant to modelers and experimentalists alike that are interested in investigating the combined effect of distinct plasticity rules in neural circuits.*

Response:

Thank you for pointing this out. We extended our discussion section to discuss potential extensions and influences of postsynaptic activity events like back-propagating action potentials.

[Manuscript, page 9-10, line 502-521:]

”Furthermore, changes in the thickness of the spine neck after LTP induction (Kruijssen and Wierenga, 2019) can modulate the flux of Ca^{2+} between spine and dendrite, influencing how strongly a synapse can be affected by heterosynaptic plasticity. Similarly, the positioning of receptors or the spine apparatus can modulate the flux of Ca^{2+} between spine and dendritic segment (Breit et al., 2018; Noguchi et al., 2005; Rosado et al., 2022). It is worth noting that

aging can change spine morphology and, consequently, synaptic plasticity (Bloss et al., 2011). However, in our study, we focused on early developmental stages, during which calcium is less confined to the spine head (Kruijssen and Wierenga, 2019; Lee et al., 2016).

One limitation of our model is that it does not include the influence of back-propagating action potentials (bAPs) in the learning rule. Wright et al., 2025 propose two distinct plasticity mechanisms in apical and basal dendrites, differing in their dependence on local co-activity and postsynaptic action potentials. However, their study found no evidence of precise millisecond-scale timing differences between somatic spikes and synaptic inputs in either apical or basal dendrites. This finding is consistent with previous research showing that bAPs are not universally present across neuron types and dendritic areas, likely due to variations in voltage-gated ion channels such as Na^+ and Ca^{2+} (Waters et al., 2005). Moreover, it has been established that bAPs are not a prerequisite for long-term potentiation (Golding et al., 2002), highlighting the complexity of synaptic plasticity mechanisms. To investigate this complex interplay between plasticity and bAPs, our model could be extended by such mechanisms using established mathematical descriptions of bAPs (Kornijcuk et al., 2020; Rackham et al., 2010) or dendritic spikes (Bono and Clopath, 2017; Legenstein and Maass, 2011).

Minor

Reviewer:

4. *The authors do a good job of discussing the relevance of their work to biological processes and mechanisms at the subcellular scale (e.g., ER and mitochondria buffering of calcium and its effect on plasticity). Their work would be better situated within the literature if they expanded the discussion in the last paragraph about the relationship between their work and functional properties of circuits that were studied at the level of dendrites – for example,*
- A. *Heterogeneous distribution of orientation selectivity of nearby synapses onto single neurons (related to Fig. 3)*
 - B. *Involvement of dendritic compartments in prediction error computations*
 - C. *Sequence generation (related to Fig. 5)*

Response:

We would like to thank you for your valuable input. We have extended the Discussion section accordingly, incorporating the reviewer’s suggestions as follows.

[Manuscript, page 10, line 522-549:]

”We demonstrated the influence of input location and spine arrangement on emergent plasticity patterns in Fig. S7. Our study focused on excitatory synapses, which predominantly host spines (Yuste, 2023). However, Agnes and Vogels, 2024 emphasized the importance of balancing excitatory and inhibitory synapses for stable synaptic weight profiles. Our model can be improved by incorporating distance-dependent spatial interaction strengths for proximal and distal dendrites, addressing a limitation of the abstract compartmental model used by Agnes and Vogels, 2024. Tsimring et al., 2025 also found that calcium activity plays a crucial role in shaping spine development, leading to synchronized spine activity during maturation, although with often imperfectly matched orientation and direction selectivity. Moreover, their study examined the effect of somatic output on calcium activity and spine properties via backpropagation signals Tsimring et al., 2025, a process that is not accounted for in our current model. Somashekar and Bhalla, 2025 demonstrated that a bistable switch in dendritic branches can selectively respond to ordered sequences, but their study lacked a clear learning mechanism to explain this phenomenon and associated synaptic weight changes. Predictive error computation is attributed to discrepancies between top-down and bottom-up signals, with basal and apical dendrites receiving different inputs (Keller and Mrsic-Flogel, 2018, Guerguiev et al., 2017). In general, extending our model by such discussed principles can provide a more detailed understanding about the role of dendrites in diverse types of neuronal computation.”

Reviewer:

5. *”different levels of input frequency”* \rightarrow *”different input frequencies”*

Response:

[Manuscript, page 2, line 73:] Thank you. We adapted the text accordingly.

Reviewer:

6. "are set equals zero" -- > "are set to zero" / "are equal to zeros"

Response:

[Manuscript, page 3, line 155:] Thank you. We adapted the main text accordingly.

Reviewer:

7. "We ensure that at least three non-stimulated spines are situated outside the group of stimulated spines on each side" What motivates this choice of driving synapses with correlated inputs within a "compact" region? Is this based on a specific experimental paper? If not, what happens if this assumption is relaxed?

Response:

This assumption is based on the experimental setup of Chater et al., 2022. Since we did not have access to the experimental data for the exact positions of spines, we arranged spines similar to their experimental images. Moreover, to account for variability of positioning of spines, we varied the distribution of the spines within this region. Furthermore, the spine density has been reported with a quite wide range from 1 to 3.1 spine / μm with the average of 2.0 spines / μm in CA1 pyramidal cells in rat hippocampus (Harris and Stevens, 1989) or about 20 spines / $10\mu\text{m}$ in layer II pyramidal cells in the mouse cortex (Ballesteros-Yáñez et al., 2006, but also see Megias et al., 2001). Following these experimental results, in our simulation, we placed 10 spines (7 stimulated and 3 unstimulated among them) in a 7 μm section of a branch, which gives us ~ 1.4 spines/ μm . We adapted the main text to clarify this point in the manuscript. [Manuscript, page 5, line 248-259:]

"Next, we extended our model considering a dendrite with multiple spines to match our model results to various experimental data sets (Chater et al. 2022; Oh et al. 2015; Tong et al. 2021). Note that the experimentally reported plasticity patterns after stimulation show a high degree of variety, lacking a coherent explanation. To integrate all three data sets, we modeled a 80 μm long dendrite hosting 16 randomly distributed spines that are governed by calcium-dependent synaptic plasticity (Fig. 3A). All protocols have been repeated for 20 different, random spine configurations, in which initial synaptic weights are randomly chosen representing different initial sizes and volumes of the spines in experiment. A random subset of 7 spines receives stimulation and forms in the following the group of stimulated spines (Stim, teal color). We ensure that at least three non-stimulated spines are situated outside the group of stimulated spines on each side ($U_{n_{\text{out}}}$, pastel yellow). All further spines are also non-stimulated and form the group of inner neighboring spines ($U_{n_{\text{in}}}$, coral pink). Note that the spatial distribution of spines is arranged such that a total of 10 spines, comprising 7 stimulated and 3 unstimulated ones, are confined to a 7 μm section of a dendritic branch, yielding a spine density of approximately 1.4 spines/ μm , following reported experimental values (Ballesteros-Yáñez et al., 2006; Chater et al., 2022; Harris and Stevens, 1989; Megias et al., 2001)."

Reviewer:

8. In Fig. 3B,C,D, it is not clear whether each point represents a synapse of one type or average over synapses of the same type in a given simulation. If the latter, then the authors should additionally show the variability of individual synapses. The points themselves are very hard to see. Same comments for Fig. S7 and S8.

Response:

[Manuscript, page 6:] In Fig. 3B,C and D, each point corresponds to the result of a simulation at that frequency for a specific arrangement of spines. Thus, for arrangement A and B of spines, the change of synaptic weights are calculated separately for each spine group (namely Stim, $U_{n_{\text{in}}}$ and $U_{n_{\text{out}}}$) and are shown as a point in each plot. The point size has been increased as suggested, and the corresponding captions have been adjusted accordingly in both figures.

Updated figure 3. Simulation and experimental results at different frequencies (2, 100, and 150 Hz). **A)** Arrangement of stimulated (in teal color) nearby unstimulated (in coral pink) and distant unstimulated spines (in pastel yellow). **B)-D)** Simulation results with low ($D=1 \frac{\mu m^2}{s}$) and physiological diffusion constant ($D=220 \frac{\mu m^2}{s}$) as well as experimental results. **B)** Simulation results with low input frequency as well as experimental observation (see Fig. 1 in Oh et al., 2013), **C)** simulation results at medium frequency and experimental observation (see Fig. 2. ciii in Tong et al., 2021), and **D)** simulation at high frequency as well as experimental observation (see Fig. S1 in Chater et al., 2022). Each data point corresponds to the synaptic weight change of a specific synapse type (Stim, Un_in, or Un_out) obtained after simulation at a particular frequency and with a specific spatial arrangement of spines. The standard deviation reflects the variability in synaptic weight changes obtained across different spatial arrangements of spines. **E)** Errors between simulation and experiments shown in B-D) for different diffusion constants (see synaptic weight patterns in Fig. S8). Black data points represent the total error for a specific spine arrangement across all frequencies and spine groups. Note that diffusion constants are plotted in logarithmic scale. The standard deviation represents the variability in error values obtained from different spatial arrangements of spines.

[Supplementary, page :]

Updated figure S7. Simulation results at in different diffusion constants and different frequencies (2, 100, and 150 Hz). Simulation results with low ($D=1$) and physiological diffusion constant ($D=220$). Simulation results with low input frequency (first column), medium frequency (second column), and high frequency (third column). Each data point corresponds to the synaptic weight change of a specific synapse type (Stim, Un_in, or Un_out) obtained after simulation at a particular frequency and with a specific spatial arrangement of spines. The standard deviation reflects the variability in synaptic weight changes obtained across different spatial arrangements of spines.

Reviewer #3

Reviewer:

Recommendation: revision

The authors present a model of calcium dynamics in stimulated and unstimulated spines, and a learning rule based on the dynamics of calcium concentration. They demonstrate that the model can reproduce a set of diverse experimental results; specifically, dependence of the outcome of plasticity on the pattern of stimulation.

The paper is interesting and important since the presented model is generic with respect to specific sources of calcium, which makes it a good starting point for extending it by implementing specific calcium channels and other sources and their combinations.

Response:

We would like to thank the reviewer for the positive feedback and valuable comments.

Reviewer:

1) One general concern is that the model introduces excessive capability of synapses/spines for plasticity - with (almost?) any single spike leading to plasticity. What are than parameters for a stable regime? More specifically, what would be a set of parameters with which spines change after strong stimulation, but remain stable during background/ "working" stimulation?

Response:

Thank you for point this out. In comments 1 and 2 Reviewer #1 raised a similar point. Therefore, we provide in the following the same response:

First, we would like to clarify that the analyses in Fig. 3 were conducted using multi-spine stimulation, in line with experimental observations showing that significant synaptic weight changes typically require the activation of more than one spine. In contrast, to better visualize the influence of spike timing, we used lower potentiation and depression thresholds in Fig. 2. We have adapted the manuscript accordingly to make this point clearer in the discussion of Fig. 2.

In addition, following the reviewer's suggestion, we simulated single-spine stimulation using the same parameters ($\theta_{d/p}$, $\gamma_{d/p}$) calibrated in Fig. 3. Please see Fig. 2. Under these conditions, the weight of neighboring spines remained unchanged, indicating stable synaptic weights. At the stimulated spine, homosynaptic plasticity yield only modest changes of about 1%. These results confirm that in our setup matching experimental results, single-spine stimulation does not result in unrealistic potentiation or depression patterns, consistent with experimental findings (Chater et al., 2022; Lee et al., 2016).

Figure 2. Synaptic weight changes after noisy inputs. Random stimulation of the 'Stim' group with 0.1 Hz results in no significant synaptic weight changes at stimulated and neighboring spines.

[Manuscript, pages 3-5:]

[...] "We initially stimulated the first spine (blue), but different to the previous protocol, here, this stimulation is followed by a stimulation of the second spine (orange) 15ms later. Compared to the first protocol, stimulation of the second spine completely changes the pattern of plasticity, triggering synaptic depression at spine 1 and potentiation of spine 2 (Fig. 2C, right). To better illustrate the impact of spike timing on synaptic plasticity, we intentionally adjusted the potentiation and depression thresholds in Fig. 2 to be lower than those in Fig. 3, thereby amplifying the visibility of subtle timing-dependent changes."

Reviewer:

2) *In the discussion, please extend discussion of what your new model adds/shows, and how it relates to prior experimental and model studies. In the present form, discussion is mostly about possible future extensions of the model (which is good and useful, but can't be the main part of the discussion).*

Response:

Thank you for the suggestion. We have extended the discussion section to include a more detailed examination of previous experimental and computational studies, and provided a comparative discussion with our current model.

[Manuscript, page 8, line 427-448:]

"We developed a computational model of molecular diffusion dynamics in a piece of dendrite and connected spines. Using a diffusion constant similar to that of calcium, our model can match results from several experimental studies of homo- and heterosynaptic plasticity. Our model also demonstrates that, given the sensitivity of calcium influx to the timing of presynaptic input spikes, the triggered homo- and heterosynaptic plasticity can lead to complex input-spike-timing-dependent plasticity patterns. This provides a rich repertoire of calcium-based learning rules without the need for postsynaptic spiking.

Previous experimental studies have reported seemingly contradictory results (Chater and Goda, 2021; Oh et al., 2015; Tong et al., 2021), yet a comprehensive mechanism explaining these discrepancies has not been proposed. Chater et al., 2022 investigated structural long-term potentiation (sLTP) using glutamate uncaging, showing that the number and arrangement of stimulated spines influence plasticity outcomes. Their mathematical model incorporated two types of proteins, but focused on abstract representations of signaling processes and potentiation, without reproducing synaptic depression within stimulated clusters (Chater et al., 2024). In contrast, our framework successfully replicated this phenomenon, which was experimentally observed by Oh et al., 2015. A recent study by Tsimring et al., 2025 monitored structural and functional turnover of dendritic spines during the critical period, finding that spine retention is strongly dependent on calcium activity and highlighting the crucial role of heterosynaptic plasticity (Tsimring et al., 2025). However, their experimental design may have led to the misclassification of spines (Tsimring et al., 2025), and their focus on functional role and pre- and postsynaptic elements makes direct comparison with our model challenging.

While our objective was to develop a computational model that balances biological realism with computational practicability, it is essential to recognize the possibility of further improvements. Future improvements could incorporate more intricate and detailed calcium sources, such as calcium-dependent channels or organelles that play a role in synaptic and dendritic calcium dynamics."

Reviewer:

3) *Presentation of results. On several occasions, self-evident things are resented as results. E.g. p. 5 "If we reduce the effective diffusion constant such that heterosynaptic plasticity will have a negligible effect on the plasticity pattern, model results always show only homosynaptic potentiation and no plasticity at unstimulated spines" - this is by design property of the model;*

"It is evident that increasing γ_d enhances the level of depression, ... There are some intermediate values of γ_d and γ_p at which the temporal windows are becoming more complex." Again, these are per design properties. What is important here, is that there is a range of γ_d and γ_p values in which plasticity windows show more complex pattern; please elaborate on this.

Response:

Thank you very much. We agree to your point and have adapted the text to better clarify our findings.

[Manuscript, page 5, line 303-311:]

"Furthermore, the induction of heterosynaptic plasticity is essential for all protocols. If we reduce the effective diffusion constant such that heterosynaptic plasticity will have a negligible

effect on the plasticity pattern, model results always show only homosynaptic potentiation and no plasticity at unstimulated spines (left, Fig. 3B-D).

We repeated all protocols with different values of the diffusion constant ($D = 1, 30, 65, 140, 220, 260, 300, 350, 400, 440 \frac{\mu\text{m}^2}{\text{s}}$) and 20 random configurations each (Fig. 3E). Then, we counted the number of instances where the resulting plasticity patterns matched the plasticity pattern of the corresponding experiment for each spine group (i.e., Stim, Un_{in}, and Un_{out}, as shown in the last column of Fig. 3B, C, and D), and derived a matching error between 0 and 1. It is evident from Fig. 3E that the lowest error value corresponds to a range of diffusion constants near the calcium diffusion constant. **Although reducing the diffusion constant in our model diminishes the heterosynaptic effect by design. However, it is worth noting that this change not only impairs heterosynaptic plasticity at the corresponding frequency, but also affects the homosynaptic pattern, which no longer matches the reported experimental results. For instance, in Fig. 3B (left panel), stimulated spines undergo potentiation at $D=1$, whereas at $D=220$ they undergo depression, consistent with experimental observations. This result supports the model assumption of calcium being one of the main candidates to communicate heterosynaptic plasticity.**

[Manuscript, page 5,7, line 331-346:]

Fig. 4A shows the categorized patterns of synaptic weights at different input time differences. It is evident that increasing γ_d enhances the level of depression, thereby shifting the temporal window towards depression for all time differences (green region in Fig. 4A). Similarly, increasing the potentiation coefficient, γ_p , shifts the temporal window towards potentiation for all time differences (red region in Fig. 4A). There are intermediate values of γ_d and γ_p at which the temporal windows become more complex. **It is noteworthy that the majority of existing experimental and theoretical studies have primarily focused on pair activity of neurons (Inglebert et al., 2020; Tazerart et al., 2020) and external calcium (Inglebert et al., 2020). Consequently, the parameter values corresponding to $\theta_{p/d}$ and $\gamma_{p/d}$ cannot be directly compared to these studies. Furthermore, previous work, such as Inglebert et al., 2020, which employed parameter fitting, still suggests multiple ranges of values for some of these parameters. Other computational studies (Graupner and Brunel, 2012; Hiratani and Fukai, 2017) have also focused on pairing protocols and proposed different values for these parameters. Although we explored a wide range of $\theta_{p/d}$ and $\gamma_{p/d}$ and selected values that match different experimental observations, a fair comparison and quantification of the parameters of our model necessitate a compatible experimental setup.**

Reviewer:

4) **Fig. 3** Axes legends are hardly visible. Please indicate source of experimental data also in the figure legend.

Response:

Thank you for pointing this out. We have changed the font size for better visibility. The citations are now declared in the caption of figures. Please see response to Comment 8, Reviewer 1 for further changes to the figure.

Updated figure 3. Simulation and experimental results at different frequencies (2, 100, and 150 Hz). **A)** Arrangement of stimulated (in teal color) nearby unstimulated (in coral pink) and distant unstimulated spines (in pastel yellow). **B)-D)** Simulation results with low ($D=1 \frac{\mu\text{m}^2}{\text{s}}$) and physiological diffusion constant ($D=220 \frac{\mu\text{m}^2}{\text{s}}$) as well as experimental results. **B)** Simulation results with low input frequency as well as experimental observation (see Fig. 1 in Oh et al., 2013), **C)** simulation results at medium frequency and experimental observation (see Fig. 2. ciii in Tong et al., 2021), and **D)** simulation at high frequency as well as experimental observation (see Fig. S1 in Chater et al., 2022). Each data point corresponds to the synaptic weight change of a specific synapse type (Stim, Un_in, or Un_out) obtained after simulation at a particular frequency and with a specific spatial arrangement of spines. The standard deviation reflects the variability in synaptic weight changes obtained across different spatial arrangements of spines. **E)** Errors between simulation and experiments shown in B-D) for different diffusion constants (see synaptic weight patterns in Fig. S8). Black data points represent the total error for a specific spine arrangement across all frequencies and spine groups. Note that diffusion constants are plotted in logarithmic scale. The standard deviation represents the variability in error values obtained from different spatial arrangements of spines.

Reviewer:

5) **Fig 4.** *Why plasticity is called heterosynaptic, while both spines were stimulated?*

Axes legends are hardly visible. In A, Y-scale might be "Synaptic weight change" In A, expand Y-scale, e.g. 0.9 - 1.15; with the present scale most space in the plots is empty, and difference between curves is hard to see. Also, think about presenting plots in a systematic order (it looks like time windows do change systematically with a shift along X or Y-axis).

In B, three lines are difficult to distinguish; consider adding color or line-style, or some other labels.

In the stimulation schemes (all figures), consider positioning spines vertically, to show stimulation on the common time-scale.

Response:

Although both spines are stimulated, in addition to the homosynaptic plasticity that has led to the weight change of the synapse, the heterosynaptic effect has modified the magnitude and/or direction of the weight change, depending on the inter-spine distance or time window between the activities of the spines. Therefore, we talk here about heterosynaptic plasticity.

Regarding the scale, we expanded the Y-scale for better readability. Since we had plotted the curves for all pairs of parameters, we chose the original Y-scale to facilitate comparison among all curves, as some values reached beyond 1.1 or below 0.9, respectively.

A)

B)

C)

Updated figure 4. Various curves of heterosynaptic, input-timing plasticity. **A)** Emergent temporal windows after stimulation of two spines at various time differences. Both spines receive one input spike as stimulation. The coefficients γ_p and γ_d correspond to the strength of potentiation and depression, respectively. Distance between spines is $1 \mu\text{m}$. Letters P and D denote potentiation and depression, respectively. Combinations of these letters indicate the order of occurrence within the total time window from -100 to 100 ms (For calcium-dependent threshold, see Fig. S1, S3). Note that spine 1 is on the left and spine 2 is on the right side of the dendritic branch, and the timing is referenced to the input time of spine 1. **B)** Patterns of calcium-based input timing synaptic plasticity for different inter-spine distance between spine 1 and spine 2 ($1, 2$ and $3 \mu\text{m}$). Due to the symmetry between spines 1 and 2, only the synaptic weight of spine 1 is shown. **C)** Triplet-dependent heterosynaptic plasticity. Emergent plasticity patterns after stimulation of two spines at various time differences with one spine receiving two input spikes and the second spine receiving one spike with a delay. The inter-input interval is 20 ms (For other calcium-dependent parameters, see Fig. S5, S6).

Reviewer:

6) **Fig. 5.** Please show somatic responses to inwards and outwards stimulation patterns before and after learning. Also, discuss and explain these non-trivial results.

Response:

We adapted the figure as suggested and discuss now the results in more detail. Consequently, Fig. 9 from the previous version of the supplementary material has been removed.

Updated figure 5. Somatic voltage during sequence selectivity via heterosynaptic plasticity.

A) The membrane potential of the soma before learning. The blue line shows the somatic response in the absence of calcium-dependent learning, resulting in no somatic spike. The dashed line shows the spiking threshold of the soma. **B) Top:** Somatic membrane potential after learning of the inward sequence. The green curve shows the response when the inward signal is presented to the dendrite after learning, while the pink curve shows the response when the outward signal is presented. The former leads to somatic firing. **Bottom:** Similar to the **top**, but for the response of the soma after learning of the outward sequence. In this case, the neuron, which has learned the outward signal, fires only when the outward sequence is presented to the dendrite. Note that the neuron's response can be modified by changing the location of its spines or the timing of the presynaptic partner. The blue dash-line represents the threshold for somatic firing. **C)** The ratio of the somatic response to the learned sequence versus the non-learned sequence. The left bars show the ratios for the protocol in which the inward sequence was presented during the learning phase, while the right bars illustrate the ratios when the outward pattern was presented during the learning phase.

Reviewer:

7) Finally, language needs improvement. The use of lab jargon should be avoided. E.g. "input spikes at a spine" is kind of nonsense; presynaptic spikes as such do not reach the postsynapse (spines).

Response:

Thank you. We carefully adapted the text. Please note that we did not highlight language changes in the main text.

References

- Agnes, E. J., & Vogels, T. P. (2024). Co-dependent excitatory and inhibitory plasticity accounts for quick, stable and long-lasting memories in biological networks. *Nature Neuroscience*, 27(5), 964–974.
- Ballesteros-Yáñez, I., Benavides-Piccione, R., Elston, G., Yuste, R., & DeFelipe, J. (2006). Density and morphology of dendritic spines in mouse neocortex. *Neuroscience*, 138(2), 403–409.

- Bloss, E. B., Janssen, W. G., Ohm, D. T., Yuk, F. J., Wadsworth, S., Saardi, K. M., McEwen, B. S., & Morrison, J. H. (2011). Evidence for reduced experience-dependent dendritic spine plasticity in the aging prefrontal cortex. *Journal of Neuroscience*, *31*(21), 7831–7839.
- Bono, J., & Clopath, C. (2017). Modeling somatic and dendritic spike mediated plasticity at the single neuron and network level. *Nature communications*, *8*(1), 706.
- Breit, M., Kessler, M., Stepniewski, M., Vlachos, A., & Queisser, G. (2018). Spine-to-dendrite calcium modeling discloses relevance for precise positioning of ryanodine receptor-containing spine endoplasmic reticulum. *Scientific reports*, *8*(1), 15624.
- Chater, T. E., Ettl, M., Goda, Y., & Tchumatchenko, T. (2022). A quantitative rule to explain multi-spine plasticity. *bioRxiv*. <https://doi.org/10.1101/2022.07.04.498706>
- Chater, T. E., Ettl, M. F., Goda, Y., & Tchumatchenko, T. (2024). Competitive processes shape multi-synapse plasticity along dendritic segments. *Nature Communications*, *15*(1), 7572.
- Chater, T. E., & Goda, Y. (2021). My neighbour hetero—deconstructing the mechanisms underlying heterosynaptic plasticity. *Current Opinion in Neurobiology*, *67*, 106–114.
- Golding, N. L., Staff, N. P., & Spruston, N. (2002). Dendritic spikes as a mechanism for cooperative long-term potentiation. *Nature*, *418*(6895), 326–331.
- Graupner, M., & Brunel, N. (2012). Calcium-based plasticity model explains sensitivity of synaptic changes to spike pattern, rate, and dendritic location (vol 109, pg 3991, 2012). *Proceedings of the National Academy of Sciences of the United States of America*, *109*(52), 21551–21551.
- Guerguiev, J., Lillicrap, T. P., & Richards, B. A. (2017). Towards deep learning with segregated dendrites. *elife*, *6*, e22901.
- Harris, K. M., & Stevens, J. K. (1989). Dendritic spines of ca 1 pyramidal cells in the rat hippocampus: Serial electron microscopy with reference to their biophysical characteristics. *Journal of Neuroscience*, *9*(8), 2982–2997.
- Hiratani, N., & Fukai, T. (2017). Detailed dendritic excitatory/inhibitory balance through heterosynaptic spike-timing-dependent plasticity. *Journal of Neuroscience*, *37*(50), 12106–12122.
- Inglebert, Y., Aljadeff, J., Brunel, N., & Debanne, D. (2020). Synaptic plasticity rules with physiological calcium levels. *Proceedings of the National Academy of Sciences*, *117*(52), 33639–33648.
- Keller, G. B., & Mrsic-Flogel, T. D. (2018). Predictive processing: A canonical cortical computation. *Neuron*, *100*(2), 424–435.
- Kornijcuk, V., Kim, D., Kim, G., & Jeong, D. S. (2020). Simplified calcium signaling cascade for synaptic plasticity. *Neural Networks*, *123*, 38–51.
- Kruijssen, D. L., & Wierenga, C. J. (2019). Single synapse ltp: A matter of context? *Frontiers in cellular neuroscience*, *13*, 496.
- Lee, K. F., Soares, C., Thivierge, J.-P., & Béique, J.-C. (2016). Correlated synaptic inputs drive dendritic calcium amplification and cooperative plasticity during clustered synapse development. *Neuron*, *89*(4), 784–799.
- Legenstein, R., & Maass, W. (2011). Branch-specific plasticity enables self-organization of nonlinear computation in single neurons. *Journal of Neuroscience*, *31*(30), 10787–10802.
- Megias, M., Emri, Z., Freund, T., & Gulyás, A. (2001). Total number and distribution of inhibitory and excitatory synapses on hippocampal ca1 pyramidal cells. *Neuroscience*, *102*(3), 527–540.
- Noguchi, J., Matsuzaki, M., Ellis-Davies, G. C., & Kasai, H. (2005). Spine-neck geometry determines nmda receptor-dependent ca²⁺ signaling in dendrites. *Neuron*, *46*(4), 609–622.
- Oh, W. C., Hill, T. C., & Zito, K. (2013). Synapse-specific and size-dependent mechanisms of spine structural plasticity accompanying synaptic weakening. *Proceedings of the National Academy of Sciences*, *110*(4), E305–E312.
- Oh, W. C., Parajuli, L. K., & Zito, K. (2015). Heterosynaptic structural plasticity on local dendritic segments of hippocampal ca1 neurons. *Cell reports*, *10*(2), 162–169.
- Rackham, O., Tsaneva-Atanasova, K., Ganesh, A., & Mellor, J. (2010). A ca²⁺-based computational model for nmda receptor-dependent synaptic plasticity at individual post-synaptic spines in the hippocampus. *Frontiers in synaptic neuroscience*, *2*, 1356.
- Rosado, J., Bui, V. D., Haas, C. A., Beck, J., Queisser, G., & Vlachos, A. (2022). Calcium modeling of spine apparatus-containing human dendritic spines demonstrates an “all-or-nothing” communication switch between the spine head and dendrite. *PLoS Computational Biology*, *18*(4), e1010069.
- Somashekar, B. P., & Bhalla, U. S. (2025). Discriminating neural ensemble patterns through dendritic computations in randomly connected feedforward networks. *eLife*, *13*, RP100664.
- Tazerart, S., Mitchell, D. E., Miranda-Rottmann, S., & Araya, R. (2020). A spike-timing-dependent plasticity rule for dendritic spines. *Nature communications*, *11*(1), 4276.

- Tong, R., Chater, T. E., Emptage, N. J., & Goda, Y. (2021). Heterosynaptic cross-talk of pre-and post-synaptic strengths along segments of dendrites. *Cell reports*, *34*(4).
- Tsimring, K., Jenks, K. R., Cusceddu, C., Heller, G. R., Ip, J. P. K., Gjorgjieva, J., & Sur, M. (2025). Large-scale synaptic dynamics drive the reconstruction of binocular circuits in mouse visual cortex. *Nature Communications*, *16*(1), 5810.
- Waters, J., Schaefer, A., & Sakmann, B. (2005). Backpropagating action potentials in neurones: Measurement, mechanisms and potential functions. *Progress in biophysics and molecular biology*, *87*(1), 145–170.
- Wright, W. J., Hedrick, N. G., & Komiyama, T. (2025). Distinct synaptic plasticity rules operate across dendritic compartments in vivo during learning. *Science*, *388*(6744), 322–328.
- Yuste, R. (2023). *Dendritic spines*. MIT press.